# Phenoconversion of Cytochrome P450 Metabolism: A Systematic Review

**DOI:** 10.3390/jcm9092890

**Published:** 2020-09-07

**Authors:** Sylvia D. Klomp, Martijn L. Manson, Henk-Jan Guchelaar, Jesse J. Swen

**Affiliations:** 1Department of Clinical Pharmacy & Toxicology, Leiden University Medical Center, 2333 ZA Leiden, The Netherlands; s.d.klomp@lumc.nl (S.D.K.); h.j.guchelaar@lumc.nl (H.-J.G.); 2Leiden Network for Personalised Therapeutics, Leiden University Medical Center, 2333 ZA Leiden, The Netherlands; m.l.manson@lacdr.leidenuniv.nl; 3Division of BioTherapeutics, Leiden Academic Centre for Drug Research (LACDR), Leiden University, 2333 CC Leiden, The Netherlands

**Keywords:** phenoconversion, pharmacogenetics, cytochrome P450, concomitant medication, comorbidities, *CYP2D6*, *CYP2C19*, *CYP3A5*, personalized medicine

## Abstract

Phenoconversion is the mismatch between the individual’s genotype-based prediction of drug metabolism and the true capacity to metabolize drugs due to nongenetic factors. While the concept of phenoconversion has been described in narrative reviews, no systematic review is available. A systematic review was conducted to investigate factors contributing to phenoconversion and the impact on cytochrome P450 metabolism. Twenty-seven studies met the inclusion criteria and were incorporated in this review, of which 14 demonstrate phenoconversion for a specific genotype group. Phenoconversion into a lower metabolizer phenotype was reported for concomitant use of CYP450-inhibiting drugs, increasing age, cancer, and inflammation. Phenoconversion into a higher metabolizer phenotype was reported for concomitant use of CYP450 inducers and smoking. Moreover, alcohol, pregnancy, and vitamin D exposure are factors where study data suggested phenoconversion. The studies reported genotype–phenotype discrepancies, but the impact of phenoconversion on the effectiveness and toxicity in the clinical setting remains unclear. In conclusion, phenoconversion is caused by both extrinsic factors and patient- and disease-related factors. The mechanism(s) behind and the extent to which CYP450 metabolism is affected remain unexplored. If studied more comprehensively, accounting for phenoconversion may help to improve our ability to predict the individual CYP450 metabolism and personalize drug treatment.

## 1. Introduction

In the last decade, pharmacogenetics-informed drug dosing has progressively moved into clinical practice for various drug–gene combinations [1,2]. For more than 100 drug–gene pairs, guidelines to manage drug treatment based on genetic test results are available from the Dutch Pharmacogenetics Working Group (DPWG) and the Clinical Pharmacogenetics Implementation Consortium (CPIC) [3,4]. However, while there is an ever-increasing body of evidence supporting the clinical use of pharmacogenetics, the increased usage has also spawned awareness of its limitations.

DPWG and CPIC provide guidelines for genes encoding cytochrome P450 (CYP450) enzymes involved in human drug metabolism, including CYP1A2, CYP2D6, CYP2C9, CYP2C19, CYP3A4, and CYP3A5, which account for the metabolism of 70–80% of approved drugs [5]. Historically, based on observed variability in pharmacokinetics, genetic variants are categorized into five different predicted metabolizer phenotypes: normal metabolizers (NM), ultra-rapid metabolizers (UM), rapid metabolizers (RM), intermediate metabolizers (IM), and poor metabolizers (PM) [3,4].

In addition to genetics, many other factors influence drug metabolism, such as sex, age, nutritional condition, hormonal and diurnal influences, concomitant drug use, or (underlying) diseases [5]. Phenoconversion (Box 1 and Box 2) is the mismatch between the genotype-based prediction of CYP450-mediated drug metabolism and the true capacity of an individual to metabolize drugs (phenotype) due to nongenetic factors. Phenoconversion can result from the use of concomitant medication [6,7], comorbidities [7], or other factors such as smoking [8,9]. The effect of phenoconversion may differ per genotype. For example, concomitant use of an enzyme inhibitor in a genotypic *CYP2D6* IM could result in a phenotypic *CYP2D6* PM. By contrast, a *CYP2D6* PM, lacking functional CYP2D6 enzyme, may be unaffected by the concomitant use of the same enzyme inhibitor since there is no enzyme to be inhibited and, hence, remain a *CYP2D6* PM. However, in current clinical practice, genetic factors are ignored when interpreting drug–drug interactions (DDIs). Obviously, a genotype–phenotype mismatch could have significant consequences in clinical settings and may result in suboptimal treatment of patients. Moreover, phenoconversion might explain some of the conflicting results from pharmacogenetic studies and difficulties with replication of reported results [8,10].

While the concept of phenoconversion has been previously described in narrative reviews [9,11,12,13,14], no systematic review providing an overview of its impact on CYP450 metabolism is available. Therefore, we performed a systematic analysis of the factors contributing to phenoconversion and their impact in the context of CYP450 metabolism.

Box 1Example of phenoconversion due to concomitant medication.John is being treated with the CYP2D6 substrate nortriptyline (75 mg per day) for his depression. John was genotyped as *CYP2D6*1/*5* and predicted to be a *CYP2D6* intermediate metabolizer (IM) according to recent consensus recommendations [15]. Consequently, his maintenance dose was adjusted to 60% of the standard dose of nortriptyline, according to DPWG guidelines [3,4]. John has recently been diagnosed with arrhythmia and has, therefore, started therapy with the weak CYP2D6 inhibitor amiodarone [16]. Subsequently, John experienced nortriptyline-related side effects, such as tremor, hyperthermia, and tachycardia. John’s nortriptyline plasma concentrations resembled those found in a *CYP2D6* PM phenotype. If John would have been genotyped as *CYP2D6*1/*1*, which is interpreted as a *CYP2D6* NM phenotype, the drug interaction with amiodarone would have had a minimal or no effect on John’s CYP2D6 phenotype, and he would have remained a *CYP2D6* NM or shifted to a *CYP2D6* IM. This case shows that the effect of drug–drug interactions (DDI) on cytochrome p450 activity is highly dependent on the patient’s genotype.

Box 2Example of phenoconversion due to comorbidity.Suzan was treated with the CYP3A5 substrate tacrolimus for immunosuppression after kidney transplantation. She was genotyped as *CYP3A5*1/*3*, which is interpreted as a *CYP3A5* IM phenotype [17]. Suzan has been successfully treated with 5 mg tacrolimus b.i.d. since her kidney transplant. However, she recently acquired an infection that caused the release of proinflammatory cytokines and increased levels of c-reactive protein (CPR) to 200 mg/L. Suzan began to experience tacrolimus-related size effects such as tremor, vomiting, and increased serum creatinine concentrations. Suzan’s tacrolimus plasma concentrations were found to be above the therapeutic range, which indicates the occurrence of phenoconversion from a *CYP3A5* IM to PM.

## 2. Methods

### 2.1. Identification of Eligible Studies

For this systematic review, identification and selection of studies were performed according to the PICO method [18]. Preferred Reporting Items for Systematic Reviews and Meta-Analyses (PRISMA) guidelines were used to prepare the report [19]. PubMed was used to identify and extract all relevant papers until June 2020. Search terms consisted of phenoconversion and synonyms or explanations of this term, together with terms describing comedication, comorbidity, and other factors that were found to be related to phenoconversion, combined with terms for the different names of CYP450 enzymes. The full search string is provided in Appendix A. Reference lists from reviews were manually checked to identify relevant cross-references. Identified records were screened on title and abstract. Comments, editorials, narrative reviews, letters (without original data), abstracts, and publications in languages other than English were excluded.

The remaining records were assessed on their appropriateness for demonstrating phenoconversion, i.e., the mismatch between the genotype-based prediction of CYP450-mediated drug metabolism and the true capacity of an individual to metabolize drugs (phenotype) due to nongenetic factors. Criteria for inclusion were (1) study of the effect of one or more nongenetic factor(s) on CYP450-mediated drug metabolism, (2) availability of relevant CYP450 genotypes resulting in a predicted metabolizer phenotype, and (3) a quantitative assessment of the true CYP450 metabolizer phenotype. The impact of phenoconversion is hypothesized to differ amongst different genotypes. Therefore, studies that solely examined the consequences of comedications for single genotypes were excluded from this review.

### 2.2. Data Extraction

For the structure of this review, phenoconversion is divided into two topics: assessing the effect of (1) concomitant drugs and other extrinsic factors such as smoking, and (2) patient- and disease-related factors such as comorbidity and age. Phenotype assignment was performed according to the latest guidelines [15,20]. For example, if the original paper used the term extensive metabolizers (EM), subjects were assigned NM. Heterozygous NMs (one wild-type allele and one reduced function) were grouped as IMs.

## 3. Results

### 3.1. Study Selection

A total of 1166 papers were identified. After screening titles and abstracts, 75 comments, editorials, letters, and reviews and 27 non-English written publications were excluded (Figure 1). A further 736 publications were excluded because they did not meet one or more of our three inclusion criteria (see Methods). Furthermore, 15 publications describing the phenoconversion of non-CYP450 drug metabolism and 93 papers without human (material) data (non-human in-vitro work and modeling studies) were excluded. Of the remaining 219 papers, 199 papers were excluded for lack of relevance after full-text evaluation. Manual checking of references identified seven additional papers. In total, 27 studies were available for analysis.

### 3.2. Impact of Studies

The 27 studies were divided into 2 groups. Table 1 shows the results for studies investigating the impact of phenoconversion on CYP450 metabolism that results from the use of concomitant drugs and other extrinsic factors. Table 2 shows the results of studies investigating the impact of patient- and disease-related factors on phenoconversion. Two classes of studies could be distinguished: Type 1 studies (n = 15) were specifically designed to study phenoconversion; Type 2 studies (n = 12) were not specifically designed to study phenoconversion and did not categorize patients into PM, IM, NM, RM, or UM phenotype. Typically, these studies showed a change in phenotype, i.e., increased plasma levels as a marker of a decrease in CYP2D6 activity, without assigning a CYP2D6 metabolizer phenotype.

### 3.3. Phenoconversion of CYP450 by Concomitant Drugs and Other Extrinsic Factors

CYP450 enzymes are involved in the metabolism of 70% to 80% of all marketed drugs [5]. The use of concomitant medication and other extrinsic factors, such as alcohol consumption, smoking, and vitamin D exposure, have all been suggested to influence the occurrence of phenoconversion of CYP450 metabolism and will be described in the next section.

#### 3.3.1. Anticonvulsants

For anticonvulsant drugs, four papers evaluating phenoconversion were identified, three related to comedication-induced phenoconversion, and one related to phenoconversion resulting from alcohol consumption. Stiripentol is a strong inhibitor of CYP1A2, CYP2D6, CYP2C9, CYP2C19, and CYP3A4 [45]. A retrospective study of 28 patients with Dravet syndrome receiving valproate and concomitant stiripentol was undertaken to elucidate the mechanism underlying the increase in serum valproate concentration produced by concomitant stiripentol therapy [21]. It was found that the increase in valproate serum concentration was larger for *CYP2C19* NM subjects compared to *CYP2C19* PMs, suggesting that *CYP2C19* NM subjects are more susceptible to the effects of CYP2C19 inhibition. In a retrospective study of 238 Japanese patients with epilepsy receiving clobazam and concomitant treatment with other antiepileptic drugs, the aim was to identify factors influencing the *N*-desmethyl-clobazam serum concentration [22]. Clobazam is mainly metabolized by CYP3A4 and partially by CYP2C19 and CYP2B6 into the active metabolite *N*-desmethyl-clobazam [46]. This metabolite is subsequently converted by CYP2C19 into the inactive metabolite 4′-hydroxynorclobazam. It was shown that concomitant treatment with the CYP450-inducing drugs phenytoin and carbamazepine resulted in a reduced concentration–dose ratio of clobazam and an increased ratio of *N*-desmethyl-clobazam for *CYP2C19* NM and *CYP2C19* IM subjects. However, within the *CYP2C19* PM group, only the clobazam ratio decreased and not the ratio of *N*-desmethyl-clobazam, which requires CYP2C19 for further metabolism. The concomitant treatment of zonisamide (a CYP2C19 substrate) and stiripentol (a strong CYP2C19 inhibitor) elevated the concentration–dose ratio of *N*-desmethyl-clobazam for *CYP2C19* NM and *CYP2C19* IM subjects. In contrast, the ratio of the *CYP2C19* PM group was unchanged, indicating phenoconversion of *CYP2C19* NMs and *CYP2C19* IMs. In a study using liver microsomes from 114 organ transplants, the donors’ demographic data, drug use, and pathological conditions were explored as potential causes of discrepancies in the genotype-based prediction of CYP2C19 activity [8]. CYP2C19 activity was assessed with mephenytoin and compared with the CYP2C19 genotype. It was found that based on CYP2C19 genetics, the CYP2C19 metabolizer phenotype was overestimated in 47% of the individuals. Chronic alcohol consumption was found to be a factor reducing CYP2C19 activity in 13% of subjects with overestimated CYP2C19 activity [8].

#### 3.3.2. Anticoagulants

One paper evaluating phenoconversion related to comedication in patients taking anticoagulant drugs was identified. In a prospective study of 82 heart surgery patients (*CYP2C9*1/*1* and *VKORC1* T/T), the influence of the CYP2C19 genotype on the occurrence of bleeding events during treatment with warfarin was investigated [23]. Half of the patients (n = 41) were concomitantly treated with lansoprazole (a CYP2C19 inhibitor) and the other half (n = 41) with rabeprazole (control group). Warfarin dose prior to concomitant treatment with a proton-pump inhibitor was based on their international normalized ratio (INR). The outcome of the study was the number of bleeding events, such as tarry stool, bleeding from colon diverticulum, conjunctival bleeding, or bleeding into the shoulder joint or surgical sites. The study found that during the 2-month follow-up, 24% of patients treated with the CYP2C19 inhibitor lansoprazole experienced bleeding events postsurgery compared to patients treated with rabeprazole (0%), which does not affect CYP2C19. Further examination of the outcomes in lansoprazole-treated patients based on the patients’ predicted phenotype revealed that 40% of the *CYP2C19* IM patients experienced a bleeding event. Comparatively, only 12% and 22% of *CYP2C19* NM and PM patients, respectively, experienced bleeding events, thus indicating that CYP2C19 inhibition by lansoprazole has a phenotype-mediated effect in patients treated with warfarin.

#### 3.3.3. Antihypertensives

One paper evaluating comedication-related phenoconversion in patients taking antihypertensive drugs was included. In a prospective study of 16 male psychiatric patients treated with thioridazine, the influence of thioridazine withdrawal on the debrisoquine (a CYP2D6 probe drug) MR was evaluated, together with the effect of the CYP2D6 genotype [24]. When treated with thioridazine (150 mg/d or higher), fourteen patients (87.5%) were phenotypical *CYP2D6* PM subjects (MR > 12.6). After complete withdrawal of thioridazine in ten patients, only two genotypical *CYP2D6* PM patients remained phenotypical *CYP2D6* PM. All of the *CYP2D6* NM genotype patients returned to *CYP2D6* phenotypical NMs after complete withdrawal. At 50 and 100 mg of thioridazine, the proportion of phenotypical *CYP2D6* NM subjects decreased to 33% and 29%, respectively. Genotypic *CYP2D6* IM patients receiving 50 mg of thioridazine per day were all converted to the *CYP2D6* PM metabolizer phenotype, compared to 67% of the genotypic *CYP2D6* NM patients. These results show that genotypic *CYP2D6* IMs are more susceptible to phenoconversion by thioridazine compared to *CYP2D6* NM. Moreover, these results indicate that concomitant use of thioridazine results in phenoconversion with a dose-dependent effect.

#### 3.3.4. Antimuscarinics

One study evaluated phenoconversion during treatment with the antimuscarinic drug tolterodine. In a prospective study of 9 depressed patients, the influence of concomitant treatment with the CYP2D6 inhibitor fluoxetine on the oral clearance of tolterodine was investigated [25]. The authors reported that the oral clearance of tolterodine decreased by 80% in *CYP2D6* NM subjects and by 93% in *CYP2D6* IM subjects. Even though no formal classification was given of the observed CYP2D6 metabolizer phenotypes, the data suggest that *CYP2D6* IMs were more frequently converted to the *CYP2D6* PM metabolizer phenotype compared to *CYP2D6* NMs.

#### 3.3.5. Antipsychotics

For antipsychotics, four papers evaluating phenoconversion were included: three attributed phenoconversion to comedication, and one investigated smoking-related phenoconversion. In a prospective study of 82 psychiatric patients receiving treatment with one of the CYP2D6 substrates aripiprazole, haloperidol, paliperidone, risperidone or zuclopenthixol, the concomitant use of a CYP2D6 inhibitor was investigated [26]. The study aimed to evaluate the impact of CYP2D6 polymorphisms on antipsychotic concentration accounting for comedication with CYP2D6 inhibitors. Phenoconversion was found to be the limiting factor for predicting an accurate dose based on CYP2D6 genotype. With the exception of paliperidone, patients concomitantly treated with a CYP2D6 inhibitor had increased antipsychotic drug concentrations. Eight out of 82 patients showed a different CYP2D6 metabolizer phenotype from that predicted based on their CYP2D6 genotype. Five genotypically predicted *CYP2D6* NM patients were converted to phenotypical *CYP2D6* PMs, and one genotypically predicted *CYP2D6* IM was a phenotypical *CYP2D6* PM. Moreover, 3 genotype-predicted *CYP2D6* NMs were phenotypical *CYP2D6* IMs. In another prospective study of 93 psychiatric patients treated with the atypical antipsychotic aripiprazole, the effect of concomitant treatment with CYP2D6 inhibitors, aripiprazole plasma concentrations, was evaluated [27]. It was reported that the increase in aripiprazole concentration depended on the CYP2D6 genotype-based prediction of the metabolizer phenotype. Aripiprazole concentrations increased, with 50% for *CYP2D6* NM subjects and 20% for *CYP2D6* UMs, resulting in an increased risk of moderate and severe side effects.

The interaction between smoking and CYP1A2 is well-established [47]. Diluted smoke condensates bind to and activate the aryl hydrocarbon receptor. Once activated, the aryl hydrocarbon receptor induces the expression of the CYP1A1 and CYP1A2 enzymes [48]. However, the magnitude of this interaction may depend on the CYP1A2 genotype. Consequently, it was hypothesized that smokers with a specific *CYP1A2*1F*-allele may require higher doses than nonsmokers of drugs metabolized by CYP1A1 and CYP1A2. In a prospective study of 37 patients with schizophrenia or schizoaffective disorder treated with olanzapine, the influence of smoking and CYP1A2 polymorphisms on the disposition of olanzapine and its metabolite 4′-*N*-desmethylolanzapine in serum and cerebrospinal fluid was investigated [28]. The authors reported a 132% and 107% increase of the 4′-*N*-desmethylolanzapine/olanzapine ratio in homozygous *CYP1A2*1F* smokers compared to smokers with other CYP1A2 polymorphisms or nonsmokers that are homozygous carriers of *CYP1A2*1F*. The increase in the metabolite to parent ratio indicates reduced CYP1A2 enzyme activity in *CYP1A2*1F* smokers. This study suggests that phenoconversion by smoking is strongly influenced by the CYP1A2 genotype; as a consequence, *CYP1A2*1F* smokers may require higher doses of olanzapine.

#### 3.3.6. Antitussive and Pain Medication

Two studies evaluated phenoconversion during antitussive and pain medication usage. In a prospective crossover study of 17 healthy volunteers, the effects of concomitant treatment with a moderate CYP2D6 inhibitor (the antidepressant duloxetine) or a strong CYP2D6 inhibitor (paroxetine) on the plasma levels of the CYP2D6 substrates tramadol and dextromethorphan were evaluated [29]. The study aimed to assess the risk of CYP2D6 phenoconversion for *CYP2D6* NMs when administered with a moderate or strong CYP2D6 inhibitor. The authors reported the difference in tramadol and dextromethorphan plasma concentrations between homozygous (*CYP2D6*1/*1*) and heterozygous (*CYP2D6*1/*17*) NM subjects. It was found that 71% of the heterozygous *CYP2D6* NMs and 25% of the homozygous *CYP2D6* NMs were converted to phenotypically *CYP2D6* IMs by duloxetine. For paroxetine, 94% of the heterozygous *CYP2D6* NMs and 56% of the homozygous *CYP2D6* NMs were converted to phenotypical *CYP2D6* PMs. Nearly all of the heterozygous *CYP2D6* NM subjects were phenoconverted with concomitant treatment of paroxetine, while 71% of subjects were phenoconverted with duloxetine. Interestingly, in homozygous *CYP2D6* IM subjects, 56% were phenoconverted when treated with paroxetine, but only 25% were phenoconverted when treated with duloxetine. These results show that the strength of the inhibitor (weak/moderate/strong) is important in the influence of phenoconversion, along with genotype. In a follow-up in-vitro study using human liver microsomes, the authors showed that the phenoconversion is likely to be explained through a difference in the amount of functional CYP2D6 enzyme and not by differences in the potency of the CYP2D6 inhibitors for the different genotypes [49]. In a prospective study of 14 Japanese schizophrenia patients, the influence of comedication with neuroleptics metabolized by CYP2D6 was investigated [6]. It was reported that concomitant administration of levomepromazine and biperiden result in higher chlorpromazine levels. Out of the 14 studied patients, 3 *CYP2D6* NM patients were converted into *CYP2D6* PMs and 2 *CYP2D6* NMs into *CYP2D6* IMs. The other 9 patients remained *CYP2D6* IMs.

#### 3.3.7. Cardiac Drugs

One paper examining phenoconversion in patients taking cardiac drugs was included. In a prospective study, 143 supraventricular tachyarrhythmia patients were concomitantly treated with flecainide and bepridil, a CYP2D6 inhibitor [50]. The objective of the study was to investigate the effects of the CYP2D6 genotype and CYP2D6 inhibitor usage on the serum S-to-R flecainide ratio. Regardless of the genotype status, all 17 concomitant medicated patients had a lower flecainide serum level, resulting in the conversion of *CYP2D6* NMs to *CYP2D6* IMs or *CYP2D6* PMs (*CYP2D6* IMs and *CYP2D6* PMs were grouped together).

#### 3.3.8. Non-Steroidal Anti-Inflammatory Drugs (NSAIDs)

One study examined phenoconversion by concomitant medication in patients using NSAIDs. In a prospective crossover study of 22 healthy volunteers, the consequences of 7-day concomitant treatment with flurbiprofen (a CYP2C9 substrate) and fluconazole (a CYP2C9 inhibitor) was investigated [30]. Coadministration of 200 mg fluconazole increased the area under the curve (AUC) of flurbiprofen in predicted *CYP2C9* NM (n = 11) to an AUC which is comparable with the AUC of *CYP2C9* IM subjects (n = 8), and a dose of 400 mg fluconazole increased the AUC even further compared to *CYP2C9* PMs (n = 2). In predicted *CYP2C9* IM subjects, both dosages of fluconazole increased the AUC of flurbiprofen, comparable with the AUCs of *CYP2C9* PM subjects. These results indicate that the concomitant use of fluconazole can cause phenoconversion in a dose- and genotype-dependent manner.

#### 3.3.9. Proton Pump Inhibitors

A study investigating phenoconversion in patients treated with proton pump inhibitors was included in this review. Fluvoxamine is an inhibitor of CYP1A2, CYP2C19, and CYP3A4. In a prospective study of 18 healthy CYP2C19 genotyped volunteers, the effect of fluvoxamine (inhibitor of CYP1A2, CYP2C19, and CYP3A4) on the pharmacokinetics of lansoprazole (CYP2C19 substrate) was investigated. Subjects received lansoprazole with and without concomitant treatment with fluvoxamine [31]. With concomitant fluvoxamine treatment, CYP2C19 inhibition was reported to be genotype-dependent, with a 2.2- and 1.9-fold decrease for *CYP2C19* NM and IM, respectively, compared to administration of lansoprazole alone. This suggests the possible phenoconversion to a lower CYP2C19 metabolizer phenotype (i.e., *CYP2C19* IM conversion to IM or PM and IM conversion to PM). There was no significant difference in lansoprazole concentration in *CYP2C19* PMs after concomitant fluvoxamine or placebo treatment.

#### 3.3.10. Antiestrogenic Drugs

Two papers investigated phenoconversion in patients using antiestrogenic drugs: one paper due to concomitant medication, and one paper due to vitamin D exposure. In a prospective cohort study, 158 breast cancer patients were treated with tamoxifen and received concomitant treatment with CYP2D6 inhibitors [32]. The study’s aim was to show that concomitant treatment with CYP2D6 inhibitors reduces the endoxifen plasma concentration. Out of the 158 patients, 17 patients used potent CYP2D6 inhibitors (paroxetine and fluoxetine), and 25 patients used weak CYP2D6 inhibitors (sertraline, citalopram, celecoxib, diphenhydramine, and chlorpheniramine). Eighty-five patients not taking any concomitant CYP2D6 medication were used as a control group. With the exception of *CYP2D6* UMs, all patients receiving concomitant treatment with a strong CYP2D6 inhibitor were phenoconverted to the *CYP2D6* PM metabolizer phenotype. The *CYP2D6* UM patients treated with weak or potent CYP2D6 inhibitors showed lower endoxifen plasma concentrations, comparable to the *CYP2D6* NM or IM metabolizer phenotype. Concomitant treatment with a weak CYP2D6 inhibitor resulted in patients with the *CYP2D6* NM genotype being converted to the IM metabolizer phenotype and IM patients being converted to the PM metabolizer phenotype. This, again, shows that in addition to the genotype, the potency of the inhibitor also plays an important role in the occurrence of phenoconversion.

Vitamin D levels can induce CYP3A4 expression by binding of the vitamin D receptor complexes to proximal promotor elements [51,52]. In a prospective study of 196 breast cancer patients receiving tamoxifen, plasma concentrations of tamoxifen and its metabolite endoxifen were evaluated [33]. The study’s aim was to gain a better understanding of the interpatient variation in endoxifen levels. The suggested induction of CYP3A4 by vitamin D was shown to influence tamoxifen and endoxifen levels. Endoxifen levels were ~20% lower during winter months (January–March) and ~8% higher during summer months (July–September) compared to average levels. Additionally, patients taking vitamin D supplements tended to have higher endoxifen levels compared to patients without supplements. In addition, patients with higher vitamin D levels were more likely to have tamoxifen levels within the therapeutic range.

### 3.4. Phenoconversion of CYP450 by Patient- and Disease-Related Factors

In addition to the use of extrinsic factors, patient- and disease-related factors such as age, comorbidities, and pregnancy may also affect the relationship between the genotype and the CYP450 metabolizer phenotype. For example, it is known that increased proinflammatory cytokines suppress CYP3A and CYP2C19 activity [53]. A number of diseases, such as infections, are known to show elevated cytokine levels.

#### 3.4.1. Age

One paper evaluating age-related phenoconversion was included. A prospective study that compared the CYP2C19 genotype and pharmacokinetics of omeprazole in young (21–36 years) and elderly (66–85 years) patients found that elderly *CYP2C19* NMs show a wider variance in omeprazole pharmacokinetics compared to young *CYP2C19* NMs [34]. Importantly, participants were not taking any medication that could affect CYP2C19 metabolism, which excludes the possibility of potential confounding by concomitant medication use in the elderly population. The data showed that 38% of the elderly subjects with the *CYP2C19* NM genotype and 42% of the elderly subjects with the *CYP2C19* IM genotype were phenotypically *CYP2C19* PMs. There were no phenotypical *CYP2C19* PM subjects in the genotypical *CYP2C19* NM or IM groups among the younger subjects. Moreover, overall plasma levels in elderly *CYP2C19* NM and *CYP2C19* IM subjects were closer to the *CYP2C19* PM metabolizer phenotype compared to younger subjects with *CYP2C19* NM or *CYP2C19* IM genotypes, demonstrating phenoconversion in the elderly.

#### 3.4.2. Cancer

Three papers evaluating the effect of cancer on the occurrence of phenoconversion were identified. In lung cancer patients, CYP2D6 genotype–phenotype mismatches had already been reported more than 25 years ago [54]. However, these mismatches were not recognized as phenoconversion at that time. The researchers speculated that if the mismatch could be explained by the disease state, the tumor, tumor products, or tumor treatment could modulate the expression (phenoconversion). However, another explanation could be the limitations of the applied genotyping since the used assay only interrogated a limited number of genetic variants. In a prospective study of 16 patients with advanced cancer and a *CYP2C19* NM genotype, the relationship between the CYP2C19 genotype and phenotype was studied [35]. Patients received omeprazole and omeprazole, and metabolite levels were measured. It was found that 25% of the patients had an omeprazole hydroxylation index of 1, which indicates a *CYP2C19* PM phenotype. Included patients received the same anticancer therapy and did not take CYP2C19 inhibitors, indicating conversion of *CYP2C19* NM to *CYP2C19* PM due to the cancer disease state. A similar study of 33 patients with advanced cancer aimed to replicate these results [36]. Out of the 33 patients, 30 were genotypically *CYP2C19* NM and 3 *CYP2C19* IM. Out of the 30 NM, 37% was phenoconverted into the *CYP2C19* PM metabolizer phenotype. This effect was not associated with proinflammatory cytokines or growth hormone levels. The authors report a possible association between decreased CYP2C19 activity and low BMI due to cachexia. A prospective study of 25 multiple myeloma patients compared the occurrence of phenoconversion in hematological malignancies and solid tumors [37]. Patients were treated with 200 mg proguanil, and CYP2C19 genotype and proguanil and cycloguanil levels were analyzed. Based on genotyping, no *CYP2C19* PMs were predicted for this cohort. However, based on proguanil metabolism, 27% of the genotypically predicted *CYP2C19* NMs and 53% of the genotypically predicted *CYP2C19* IMs were phenotypically *CYP2C19* PMs.

#### 3.4.3. Inflammation

Six studies evaluated phenoconversion as a result of inflammation. In a prospective study of 52 patients with Behcet’s disease and 96 healthy volunteers, the influence of Behcet’s disease and CYP2C9 genotype on the activity of CYP2C9 (phenotypical determined by measuring the metabolic ratio of losartan) was determine [38]. The 31 patients that were genotypically classified as *CYP2C9* NM had a mean metabolic ratio that was comparable to the observed metabolic ratio of losartan in *CYP2C9* IM healthy volunteers. These results indicate that Behçet’s disease, a systemic inflammatory disorder of the blood vessels, can cause phenoconversion. In a prospective study of 31 patients with hepatitis C virus (HCV)-positive chronic hepatitis or cirrhosis and 30 healthy volunteers, the interaction between chronic liver disease and the CYP2C19 genotype was assessed [39]. Metabolic ratios of omeprazole/5-hydroxy omeprazole were used as the phenotypic test of CYP2C19 activity. In healthy volunteers, metabolic ratios of 0.81, 1.55, and 15.5 were observed in *CYP2C19* NM, IM, and PM, respectively. In contrast, patients with chronic liver disease, genotypically classified as *CYP2C19* NM, IM, and PM, displayed metabolic ratios of 17.15 (21.1-fold change), 20.02 (12.4-fold change), and 26.04 (1.9-fold change), respectively, which would phenotypically classify them all as CYP2C19 poor metabolizers. These results demonstrate that chronic liver disease resulting from HCV infection can reduce CYP2C19 enzymatic activity and cause phenoconversion in an infection- and genotype-dependent manner. A prospective study of 36 patients treated with intravenous or oral voriconazole aimed to investigate the effect of inflammation on the voriconazole metabolic ratio. It was found that for *CYP2C19* UMs, NMs, and IMs, at higher C-reactive protein (CRP) concentrations, the metabolic ratio was decreased, and, consequently, the voriconazole trough concentration was increased [40]. The metabolic ratio was decreased most for *CYP2C19* UM subjects and the least for *CYP2C19* IMs, suggesting that the degree of phenoconversion is influenced by its genotype. In a study using microsomes from 114 organ transplants described above (see alcohol consumption), the effect of inflammation on phenoconversion was investigated [8]. Diseases with inflammatory processes such as rheumatoid arthritis and gastrointestinal perforation were found to be associated with a reduction of CYP2C19 activity in 47% of the study population. The effect of inflammation on the phenoconversion of CYP2D6 was investigated in a prospective study in 1723 patients with a chronic HCV infection [41]. HCV infection is associated with the presence of liver kidney microsomal type 1 (LKM-1) antibodies, which are directed against the body’s endogenous CYP2D6. It was hypothesized that the LKM-1 antibodies could be a factor influencing the CYP2D6 genotype–phenotype mismatch. When LKM-1-negative patients were compared to LKM-1-positive patients, up to six-fold reduction of CYP2D6 metabolic activity was found in patients with a high level of LKM-1 antibodies (the positive patients). The metabolizer phenotype was explained by genotype in 3 out of the 10 LKM-1-positive subjects. For the others, six *CYP2D6* NM subjects were converted to the *CYP2D6* IM metabolizer phenotype, and one *CYP2D6* NM was converted to the *CYP2D6* PM metabolizer phenotype, showing phenoconversion in 70% of the cases. In a prospective study of 63 stable kidney transplant recipients, CYP3A phenoconversion was studied. The CYP3A5 genotype was determined, and immunosuppressant levels were measured [42]. The authors reported that 10 *CYP3A5*1* recipients had a lower CYP3A5 activity than expected based on the CYP3A5 genotype, indicating phenoconversion. It was found that higher indoxyl sulfate plasma concentrations were associated with phenoconversion of CYP3A5. While the exact mechanism remains unknown, it is recognized that indoxyl sulfate upregulates the activity of nuclear factor κΒ (the most important inflammatory transcription factor), and nuclear factor κΒ decreases histone 4 acetylation in the CYP3A promotor, which downregulates CYP3A expression [55,56]. Consequently, in patients with the *CYP3A5*1* allele and high blood indoxyl sulfate, the dose may need to be adjusted for drugs metabolized by CYP3A [42].

#### 3.4.4. Pregnancy

For pregnancy, two papers assessing phenoconversion were included. In a prospective study of 140 women, the effect of pregnancy on CYP2D6 activity was investigated [43]. To assess CYP2D6 activity, dextromethorphan/dextrorphan metabolic ratios were measured during and after pregnancy. It was reported that 29% and 63% of subjects with a *CYP2D6* NM and *CYP2D6* IM genotype showed a decreased metabolic ratio, indicating increased CYP2D6 activity. By contrast, *CYP2D6* PMs showed an increased metabolic ratio during pregnancy. These observations show that CYP2D6 activity may be induced in *CYP2D6* NM and IM but not in *CYP2D6* PM subjects during pregnancy. In another prospective study of 74 pregnant women receiving the antidepressant paroxetine, comparable results were reported [44]. CYP2D6 enzymatic activity was increased in genotypic subjects with *CYP2D6* NM and UM genotypes and reduced in subjects with a *CYP2D6* PM genotype. For *CYP2D6* IMs, it was found that the paroxetine plasma concentration is not affected by pregnancy. The authors suggest that the increase of paroxetine plasma levels in genotypic *CYP2D6* PMs can be explained by a decreased activity of other CYP450 enzymes involved in paroxetine metabolism. Interestingly, the two studies investigating the effect of pregnancy on phenoconversion reported conflicting results for *CYP2D6* IMs and *CYP2D6* PMs. A potential explanation could be that the drug studies are not specific probes for CYP2D6 and are metabolized by additional CYP450 enzymes that have reduced activity during pregnancy. For *CYP2D6* PMs, the result is an altered metabolic capacity; for *CYP2D6* IM subjects, both processes (induced and reduced metabolic capacity) will middle out, resulting in an unaffected plasma concentration [44].

In summary, the phenoconversion of CYP450 due to extrinsic factors has been mostly studied for the use of concomitant medication. In addition, the use of alcohol, smoking, and vitamin D exposure have been described. Out of the 14 studies, six studies formally concluded that phenoconversion occurred. The most frequently described effect was the conversion of the NM metabolizer phenotype to the PM metabolizer phenotype because of the concomitant use of CYP2C19 or CYP2D6 inhibitors. The studies show that inhibition of the different CYP450 enzymes results in lower enzyme activity and, consequently, in higher drug levels of the substrates (Figure 2). By contrast, the use of a CYP450 inducer such as carbamazepine or smoking results in the gain of function of CYP450 enzyme activity (Figure 2). The occurrence of phenoconversion due to patient- and disease-related factors has received very limited attention to date. Only four factors of phenoconversion, age, cancer, inflammation, and pregnancy, have been studied. Nevertheless, this resulted in 12 studies, of which eight reported phenoconversion occurring in ~30% of the patients. The studied disease states included cancer and inflammation, resulting in physiological changes that include, for example, nutrition or higher cytokine levels. These studies provide evidence supporting the hypothesis that higher cytokine levels result in decreased CYP450 activity, resulting in lower drug metabolism and higher drug plasma (Figure 2). The studied patient-related factors are age and pregnancy; age resulted in phenoconversion into a lower metabolizer phenotype due to decreased CYP450 enzyme activity (Figure 2). In conclusion, concomitant medication and distinct patient- and disease-related factors have been shown to modulate the activity of the critical CYP450 enzymes, which resulted in changes in drug metabolism that could not have been predicted from its genotype nor by simply reviewing it as a traditional drug–drug interaction.

## 4. Discussion

In this systematic review, we assessed studies that investigated phenoconversion by concomitant drugs and other extrinsic factors and patient- and disease-related factors. Out of the 27 identified studies, 10 studies demonstrate phenoconversion for a specific genotype group. Phenoconversion into a lower metabolizer phenotype was reported for concomitant use of CYP450-inhibiting drugs, increasing age, cancer, and inflammation. Phenoconversion into a higher metabolizer phenotype was reported for CYP450 inducers and smoking. Moreover, alcohol consumption, pregnancy, and vitamin D exposure are factors where the study data suggested phenoconversion. The interplay between genetic and nongenetic factors that results in phenoconversion should, therefore, be considered more often.

The use of concomitant medication is frequently reported as a source of phenoconversion. This finding is not unexpected, as the impact of DDIs on drug pharmacokinetics is well established and part of the daily routine of pharmacists. Importantly, several studies included in this review show that the outcome of specific DDIs varies between different genotypes [22,23,27,29,30,31,32,57], a finding that is currently not accounted for in clinical decision-making. As expected, the concomitant administration of strong CYP2D6 or CYP2C19 inhibitors caused phenoconversion in almost all subjects. However, reported data suggest that genotypic IM may be more susceptible to phenoconversion by CYP2D6, CYP2C9, and CYP2C19 inhibitors than PMs, UMs, RMs, or NMs. This finding may be particularly important for the concomitant use of weak or moderate CYP450 inhibitors, which hypothetically will have the strongest effect on phenoconversion in IM subjects and warrant further investigations.

Another recognized source of phenoconversion of CYP450 drug metabolism are comorbidities, including cancer. Several studies included in this review established that the effect of specific drug–disease interactions on CYP450 metabolism varies between different CYP450 genotypes [35,36,37,41,42]. In contrast to phenoconversion resulting from external factors, the effects of comorbidities are not yet taken into account in the management of drug treatment. Moreover, the exact mechanisms underlying phenoconversion induced by disease states such as cancer and inflammation remain to be elucidated. Liver diseases have also been suggested as a source of phenoconversion. However, while a number of studies did show changes in plasma levels due to liver diseases, no adequate genotype assessments were available in those studies [58,59,60]. Therefore, for disease-induced phenoconversion, further studies are also necessary to help to integrate pharmacogenetic information for personalizing the control of drug–disease interactions.

Our systematic search identified only 27 eligible studies investigating phenoconversion. Most of the identified studies were not specifically designed to study phenoconversion and included retrospective analyses with small sample sizes and limited genotyping. While quantitative assessment of the true CYP450 metabolizer phenotype was performed, no formal metabolizer phenotype was assigned. The assessed studies reported large numbers of genotype–phenotype discrepancies. Nevertheless, the impact on the effectiveness and toxicity remains mostly unknown. To improve our understanding of phenoconversion studies with comprehensive CYP450 genotyping, a quantitative assessment of the true CYP450 metabolizer phenotype and, ideally, a well-defined drug–response phenotype are required.

The studies included in our review mostly consider drugs that are metabolized by 1-2 CYP450 enzymes. Moreover, the effect of a single concomitant drug or other factor is investigated. However, in reality, many drugs have a complex metabolism, and patients suffer from multiple morbidities and are treated with multiple drugs simultaneously [22,61]. A recent study of patients treated with clozapine investigated the use of genotype- and comedication-corrected scores to predict the impact on clozapine exposure [62]. It was shown that by using the scores, the number of *CYP1A2* UMs increased by 1.1-fold, *CYP2D6* IMs and PMs by 1.8-fold, and *CYP2C19* IM by 1.7-fold, respectively. In the future, this type of model that combines genetic information with information on comorbidities, concomitant drug treatment, and other external factors may help to improve our capability to predict the outcome of drug treatment.

In this review, we have focused on clinical studies investigating phenoconversion. However, other approaches, including the use of in-vitro or in-silico models, may be feasible. For in-vitro studies, this requires model(s) in which the impact of the phenoconversion factor can simultaneously be assessed for different genotypes. This can be achieved either via liver models with endogenous pharmacogenetic variation, such as primary human hepatocytes (PHHs), liver microsomes, or induced pluripotent stem cells (iPSCs) [63,64,65,66], or through hepatocytes in which pharmacogenetic variants are exogenously introduced via CRISPR [67]. Secondly, phenoconversion for some conditions (e.g., inflammation) may only develop after prolonged or repeated exposure, which requires in-vitro models, such as 3D liver spheroids, in which the long-term consequences of these factors for drug metabolism can be adequately studied [68,69]. In-silico models, such as physiologically based pharmacokinetic (PBPK) models, have also been suggested as an excellent approach to study phenoconversion [70,71,72,73,74,75]. These models are particularly suitable to investigate the effects of complex interactions between multiple drugs and disease states.

## 5. Conclusions

Phenoconversion of CYP450 metabolism is caused by both extrinsic factors, such as the use of concomitant drugs, as well as patient- and disease-related factors. The mechanism(s) behind and extent to which CYP450 metabolism is affected by these factors remain largely unexplored. If studied more comprehensively, accounting for phenoconversion may help to improve our ability to predict the individual CYP450 metabolism and personalize drug treatment.

## Figures and Tables

**Figure 1 jcm-09-02890-f001:**
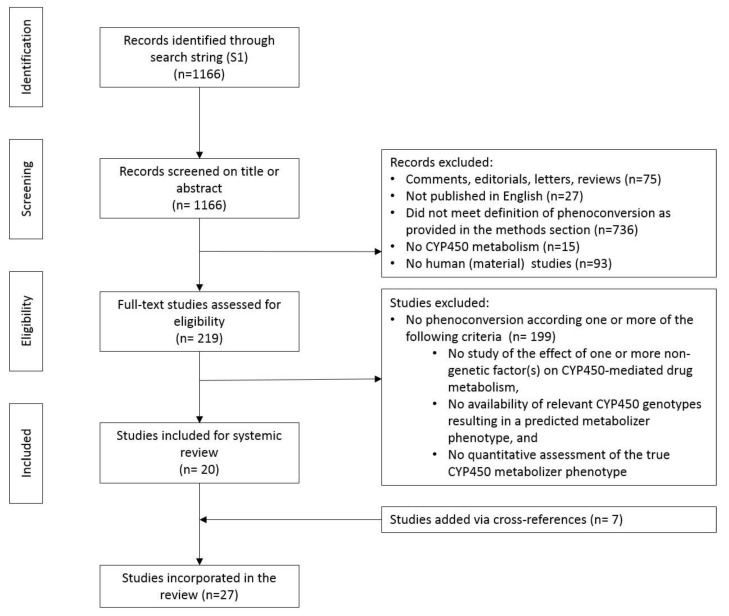
Study flow diagram of the retrieval and review process (Appendix A).

**Figure 2 jcm-09-02890-f002:**
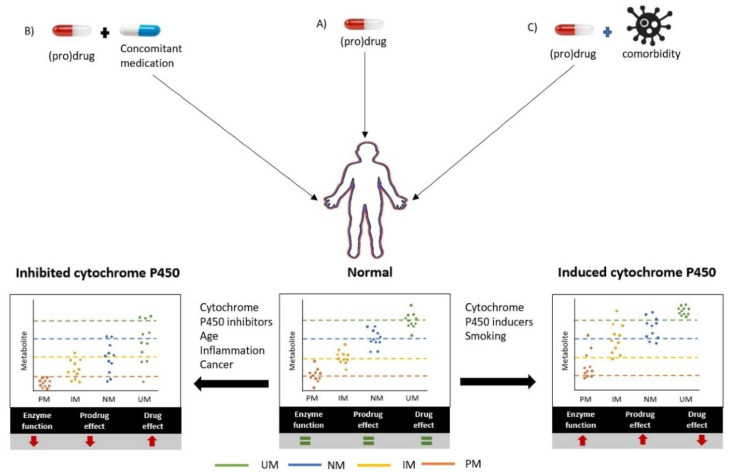
Cytochrome P450-mediated drug metabolism can be inhibited or induced by the presence of co-medication and/or comorbidities, for which the magnitude of the changes depends on the host’s genotype. The metabolite concentrations in blood are plotted for the genotype-based predicted phenotypes (UM, NM, IM, and PM). This is presented for the normal situation (**A**) and for scenarios in which (**B**) CYP450 activity is inhibited by concomitant medication or (**C**) induced by a comorbidity. The resulting effects on drug efficacy are opposing for drugs and prodrugs. UM = ultra-rapid metabolizer, NM = normal metabolizer, IM = intermediate metabolizer, PM = poor metabolizer.

**Table 1 jcm-09-02890-t001:** The impact of phenoconversion on CYP450 metabolism that results from the use of concomitant drugs and other extrinsic factors.

Pharmacological Class	Drug	Enzyme	Cause of Phenoconversion	Type of Interaction	Drug/Factor Responsible for Phenoconversion	Type of Study	Genotype Assessment	Phenotype Assessment	Effect of Phenoconversion	Subjects	Ref
Anticonvulsants	Valproate	CYP2C19	Comedication	Inhibition	Stiripentol	2	*CYP2C9*1*/**2*/**3*	Concentration-to-dose ratio of valproate	NM: Valproate serum concentration ↑↑	28	[21]
							*CYP2C19*1*/**2*/**3*		PM: Valproate serum concentration ↑	(Dravet syndrome patients)	
	Clobazam	CYP2C19	Comedication	Inhibition	Stiripentol/zonisamide	2	*CYP2C19*1*/**2*/**3*	Clobazam and *N*-desmethyl-clobazam via HPLC	22 NM zonisamide and stiripentol users (total 97 NM patients): *N*-desmethyl-clobazam concentration dose ratio ↑	238	[22]
							NM: *1/*1		25 IM zonisamide and stiripentol users (total 133 IM patients): *N*-desmethyl-clobazam concentration dose ratio ↑	(epilepsy patients)	
							IM: **1/*2* or **1/*3*		17 PM zonisamide and stiripentol users (total 72 PM patients): *N*-desmethyl-clobazam concentration dose ratio =		
							PM: **2/*2*, **2/*3* or **3/*3*				
	Clobazam	CYP2C19	Comedication	Induction	Phenytoin/carbamazepine	2	*CYP2C19*1*/**2*/**3*	Clobazam and *N*-desmethyl-clobazam via HPLC	51 NM phenytoin/carbamazepine users (total 97 NM patients): clobazam concentration dose ratio ↓	238	[22]
							NM: **1/*1*		*N*-desmethyl-clobazam concentration dose ratio ↑	(epilepsy patients)	
							IM: **1/*2* or **1/*3*		75 IM phenytoin/carbamazepine users (total 133 IM patients): clobazam concentration dose ratio ↓		
							PM: **2/*2*, **2/*3* or **3/*3*		*N*-desmethyl-clobazam concentration dose ratio ↑		
									36 PM phenytoin/carbamazepine users (total 72 PM patients): clobazam concentration dose ratio ↓		
									*N*-desmethyl-clobazam concentration dose ratio =		
	Mephenytoin	CYP2C19	Dietary	Inhibition	Alcohol consumption	2	*CYP2C19*1*/**2*/**3*/**4*/**17*	Mephenytoin 4′-hydroxylation and mephenytoin via HPLC	The genotype-predicted phenotype overestimated (47%) the measured phenotype. For the overestimated group, 7 were alcohol users, resulting in lower CYP2C19 activity.	114	[8]
							PM: two loss of function alleles (**2*, **3* or **4*)	CYP2C19 activity:		(liver microsomes)	
							IM: one functional and one loss of function allele	PM: <8 pmol/(mg protein*min)			
							NM: **1/*1*	UM: >75 pmol/(mg protein*min)			
							UM: one or two **17* alleles	IM: between 8 and 23 pmol/(mg protein*min)			
							(except **2/*17*: IM or NM are accepted)	NM: between 23 and 75 pmol/(mg protein*min)			
Anticoagulants	Warfarin	CYP2C19	Comedication	Inhibition	Lansoprazole	2	*CYP2C19* NM, IM and PM	*	There were no bleeding events in the rabeprazole control group (n = 41). Percentage of bleeding events in lansoprazole group (n = 41):	82	[23]
									NM: 12% ↑	(open heart surgery patients)	
									IM: 40% ↑↑		
									PM: 22% ↑		
									Showing de interaction between warfarin, lansoprazole and CYP2C19, (DDGI)		
Antihypertensive	Debrisoquine	CYP2D6	Comedication	Inhibition	Thioridazine	1	*CYP2D6*1*/**3*/**4*/**5*/**6*	Phenotyping by debrisoquine, debrisoquine and 4-hydroxydebrisoquine ratios via GC	8 NM→PM	16	[24]
									2 PM = PM	(psychiatric patients)	
Antimuscarinics	Tolterodine	CYP2D6	Comedication	Inhibition	Fluoxetine	2	*CYP2D6*1*/**3*/**4*	Dealkyl and carboxyl tolterodine LC-ESI-MS/MS	NM: oral clearance of tolterodine ↓ (80%)	9	[25]
								Fluoxetine and norfluoxetine via HPLC-UV	IM: oral clearance of tolterodine ↓↓ (93%)	(depressed patients)	
									Possible: NM→PM		
Antipsychotics	Aripiprazole, haloperidol, risperidone, paliperidone zuclopenthixol	CYP2D6	Comedication	Inhibition	Strong inhibitors (paroxetine/bupropion)	1	*CYP2D6*1*/**2*/**3*/**4*/**5*/**6*/**7*/**8*/**9*/**10*/**11*/**15*/**17*/**29*/**41*, deletions and duplications	Concentration of parent compounds antipsychotics via HPLC-MS/MS	8 patients taking CYP2D6 inhibitors	82	[26]
					Moderate inhibitors (sertraline/duloxetine)				Strong inhibitors:	(psychiatric patients)	
									4 NM→PM		
									1 IM→PM		
									Moderate inhibitors:		
									3 NM→IM		
	Aripiprazole	CYP2D6	Comedication	Inhibition	CYP2D6 inhibitors	2	*CYP2D6*1*/**3*/**4*/**5*/**6*/**10*/**41* and duplication, *CYP3A4*1*/**1B*/**22*, and *CYP3A5*1*/**3*	Aripiprazole, dehydroaripiprazole, *N*-dealkyl-aripiprazole, monohydroxy-aripiprazole and dehydroaripiprazole via LC-MS/MS	NM: aripiprazole concentration ↑↑ (50%)	93	[27]
									UM: aripiprazole concentration ↑ (20%)	(psychiatric patients)	
	Olanzapine	CYP1A2	Smoking	Induction	Polycyclic aromatic hydrocarbons	2	*CYP2D6*1*/**3*/**4*/**5*/**6*/**41* and duplication	Olanzapine and desmethylolanzapine via LD-MS	*CYP1A2*1F/*1F* smokers: 4′-*N*-desmethylolanzapine/olanzapine ratio ↑:	37	[28]
							PM: two defective alleles (**3*, **4*, **5* or **6* )		132% compared to smokers with other CYP1A2 polymorphisms and, 107% compared to nonsmokers that are homozygous carriers of *CYP1A2*1F*	(schizophrenia or schizoaffective disorder patients)	
							IM: one defective allele				
							NM: **1/*1*				
							UM: gene duplication together with **1*				
							*ABCB1*				
							*CYP1A2*1*/**1C*/**1D*/**1K*/**1F*				
Antitussive and pain medication	Dextromethorphan, tramadol	CYP2D6	Comedication	Inhibition	Duloxetine/paroxetine	1	*CYP2D6*1*/**2*/**3*/**4*/**6*/**7*/**8*/**9*/**10*/**17*/**29*/**35*/**41*/**5* (deletion) and duplication	Phenotyping of dextromethorphan and tramadol by LC–MS/MS followed by LLE	Duloxetine:	17	[29]
								PM: UMR_DEM/DOR_ > 0.3	71% NM→IM	(healthy volunteers)	
								IM: 0.03 < UMR < 0.3	25% NM→PM		
								NM: 0.003 < UMR < 0.03	Paroxetine:		
								UM: UMR < 0.003	94% heterozygous NM→PM		
									56% homozygous NM→PM		
	Dextromethorphan	CYP2D6	Comedication	Inhibition	Levomeprazine/biperiden	1	*CYP2D6*1*/**2*/**3*/**4*/**5*/**8*/**10*/**18*/**21*	Dextromethorphan and dextrorphan via HPLC	14 patients:	104	[6]
								Levomepromazine and biperiden via GCMS	3 NM→PM	(90 healthy volunteers)	
									2 NM→IM	(14 schizophrenia patients)	
									9 stayed IM		
Cardiac drugs	Flecainide	CYP2D6	Comedication	Inhibition	Bepridil	1	*CYP2D6*1*/**2*/**4*/**5*/**10*/**14*/**21*/**36*	Flecainide enantiomers via HPLC	All 17 concomitant treated NM patients showed a serum flecainide S/R ratio of PM subjects: NM→PM	143	[22]
							NM: **1* and **2*			(supraventricular tachyarrhythmia patients)	
							IM: **10*				
							PM: **4*/**5*/**14*/**21*/**36*				
NSAID	Flurbiprofen	CYP2C9	Comedication	Inhibition	Fluconazole	1	*CYP2C9*1*/**2*/**3*	Flurbiprofen, 4′-hydroxyflurbiprofen and fluconazole via HPLC	200 mg fluconazole:	189	[30]
									11 NM→IM	(healthy subjects)	
									8 IM→PM		
									2 PM = PM		
									400 mg fluconazole:		
									11 NM→PM		
									8 IM→PM		
									2 PM = PM		
Proton pump inhibitors	Lansoprazole	CYP2C19	Comedication	Inhibition	Fluvoxamine	2	*CYP2C19*1*/**2*/**3*	Lansoprazole enantiomers and lansoprazole sulphone via HPLC	NM: (R)-lansoprazole AUC ↑↑ (903%)	18	[31]
							NM: **1/*1*		(S)-lansoprazole AUC ↑↑ (1664%)	(healthy volunteers)	
							IM: **1/*2* or **1/*3*		IM: (R)-lansoprazole AUC ↑ (462%)		
							PM: **2/*2* or **2/*3*		(S)-lansoprazole AUC ↑ (781%)		
Antiestrogens	Tamoxifen	CYP2D6	Comedication	Inhibition	Potent inhibitors: paroxetine and fluoxetine	1	*CYP2D6*1*/**3*/**4*/**6*/**7*/**8*/**10*/**11*/**14*/**15*/**17*/**19*/**20*/**25*/**26*/**29*/**31*/**35*/**36*/**40*/**41* and duplications	Tamoxifen and metabolites via HPLC	42 CYP2D6 subjects with CYP2D6 inhibitor compared to 85 subjects without concomitant medication.	158	[32]
					Weak inhibitors: sertraline, citalopram, celecoxib, diphenhydramine, and chlorpheniramine				Mean group endoxifen plasma concentrations by potent inhibitor usage:	(breast cancer patients)	
									1 UM→IM		
									5 NM→PM		
									11 IM→PM		
									Mean group endoxifen plasma concentrations by weak inhibitor usage:		
									3 UM→NM or IM		
									10 NM→IM		
									12 IM→PM		
	Tamoxifen	CYP3A4	Other	Induction	Vitamin D	2	*CYP3A4*1/*22*, *POR*28*, *CYP2C9*1/*2/*3*, *CYP2B6*1/*4*/**5*/**6*, *MDR1* c.3435C > T, *BCRP* c.421C > A/c.34G > A, *CYP3A5*1/*3*, *CYP2D6*1/*3*/**4*/**5*/**9*/**10*/**41*	Tamoxifen, NDM-tamoxifen, 4-OH-tamoxifen, Z-endoxifen, Z-3-OH tamoxifen and Z-α-OH-tamoxifen via LC-MS/MS normalized to 4-β-OH-cholesterol via UHPLC-MS/MS and 25-OH-vitamin D via ELISA	During winter months endoxifen levels ↓ (20%)	196	[33]
										(breast cancer patients)	

Type of study refers to “designed with the objective to study phenoconversion” (Type 1) or “not specifically designed to study phenoconversion, did not categorize patients into PM, IM, NM, RM or UM phenotype” (Type 2). AUC = area under the curve, LC–ESI–MS/MS = liquid chromatography electrospray ionization tandem mass spectrometric, HPLC–UV = high pressure liquid chromatography coupled ultraviolet, HPLC–MS/MS = high pressure liquid chromatography tandem mass spectrometry, LLE = liquid–liquid extraction, GCMS = gas chromatography mass spectrometry, LD–MS = laser-assisted desorption mass spectrometry, UHPLC–MS/MS = = ultrahigh pressure liquid chromatography tandem mass spectrometry, ELISA = enzyme-linked immuno sorbent assay, UMR = urinary metabolic ratio, UM = ultra-rapid metabolizer, NM = normal metabolizer, IM = intermediate metabolizer, PM = poor metabolizer, “→” = converted to, “↑” = increase, “↑↑” = larger increase, “↓” = decrease, “↓↓” = larger decrease, “=” = did not change. *This study did not perform phenotype assessment; nevertheless, the study showed the impact of phenoconversion and, therefore, the study was included in this systemic review. The impact of phenoconversion is analyzed via hemorrhagic complications (minor bleeding events such as tarry stool, bleeding from colon diverticulum, conjunctival bleeding, or bleeding into the shoulder joint or surgical sites).

**Table 2 jcm-09-02890-t002:** The impact of phenoconversion on CYP450 metabolism that results from the use of patient- and disease-related factors.

Cause of Phenoconversion	Drug	Enzyme	Result	Type of Study	Genotype Assessment	Phenotype Assessment	Effect of Phenoconversion	Subjects	Ref
Age	Omeprazole	CYP2C19	Reduced activity	1	*CYP2C19*1*/**2*/**3*	Omeprazole (10 mg for elderly and 20 mg for young), omeprazole, 5-hydroxyomeprazole and omeprazole sulfone via LC	38% NM→PM	51	[34]
							42% IM→PM	(healthy volunteers: older age	
								(elderly 66–85 years, younger 21–36))	
Cancer	Omeprazole	CYP2C19	Reduced activity	1	*CYP2C19*1*/**2*	Phenotyping via the prodrug omeprazole (20 mg) measured in plasma.	25% NM→PM	16	[35]
						NM: log10 [OM]/[OH-OM] <1		(advanced cancer patients)	
						PM: log10 [OM]/[OH-OM] ≥1			
	Omeprazole	CYP2C19	Reduced activity	1	*CYP2C19*1*/**2*/**3*	Phenotyping via the prodrug omeprazole (20 mg) measured in plasma.	37% NM→PM	33	[36]
						NM: log10 [OM]/[OH-OM] <1	(30 NM patients)	(advanced cancer patients)	
						PM: log10 [OM]/[OH-OM] ≥1			
	Proguanil	CYP2C19	Reduced activity	1	*CYP2C19*1*/**2*/**3*/**17*	Proguanil (PG) (200 mg), PG and cycloguanil (CG) via HPLC	27% NM→PM	25	[37]
						NM: PG/CG ≥ 1	53% IM→PM	(multiple myeloma patients)	
						PM: PG/CG < 1			
Inflammation	Losartan	CYP2C9	Reduced activity	1	*CYP2C9*1*/**2*/**3*	Losartan and metabolite (E-3174) via HPLC	Mean group metabolic ratio in patients:	51	[38]
							31 NM→IM	(patients with Behçet’s disease)	
							20 IM→PM	96	
								(healthy volunteers)	
	Omeprazole	CYP2C19	Reduced activity	1	*CYP2C19*1*/**2*/**3*	Omeprazole and 5--hydroxy omeprazole via LC	Mean group metabolic ratio of omeprazole/5-hydroxy omeprazole in patients:	31	[39]
					NM: **1/*1*		NM→PM	(patients with chronic hepatitis or cirrhosis that tested positive for hepatitis C virus (HCV))	
					IM: **1/*2* or **1/*3*		IM→PM	30	
					PM: **2/*2*, **2/*3* or **3/*3*		PM = PM	(healthy volunteers)	
	Voriconazole	CYP2C19	Reduced activity	2	*CYP2C19*1*/**2*/**3*/**17*	Voriconazole and voriconazole-*N*-oxide concentrations were measured via LC-MS/MS	20 patients applicable for genotyping:	36	[40]
					UM: **1/*17*		Metabolic ratio:	(patients treated with intravenous or oral voriconazole)	
					NM: **1/*1*		UM = −0.994147^N^		
					IM: **1/*2* or **1/*3*		NM = −0.991972^N^		
					PM: **2/*2*, **2/*3* or **3/*3*		IM = −0.986512^N^		
							Voriconazole trough concentration:		
							UM = +1.003685^N^		
							NM = +1.004965^N^		
							IM = +1.009365^N^		
							N = CRP concentration		
	Mephenytoin	CYP2C19	Reduced activity	2	*CYP2C19*1*/**2*/**3*/**4*/**17*	Mephenytoin 4′-hydroxylation and mephenytoin via HPLC	The genotype-predicted phenotype overestimated (47%) the measured phenotype. For the overestimated group, 9 had inflammatory diseases (rheumatoid arthritis and gastrointestinal perforation), resulting in lower CYP2C19 activity.	114	[8]
					PM: two loss of function alleles (**2*, **3* or **4*)	CYP2C19 activity:		(liver microsomes)	
					IM: one functional and one loss of function allele	PM: <8 pmol/(mg protein*min)			
					NM: **1/*1*	UM: >75 pmol/(mg protein*min)			
					UM: one or two **17* alleles	IM: between 8 and 23 pmol/(mg protein*min)			
					(except **2/*17*: IM or NM is accepted)	NM: between 23 and 75 pmol/(mg protein*min)			
	Dextromethorphan	CYP2D6	Reduced activity-	1	*CYP2D6*	Dextromethorphan (DEM) (2.5 mg), DEM and dextrorphan (DOR) via HPLC	10 LKM-1 positive subjects:	1723	[41]
					33 allelic variants (AmpliChip^®^)	UM: DEM/DOR < 0.003	6 NM→IM	(chronic hepatitis C patients)	
						NM: 0.003 < DEM/DOR < 0.03	1 NM→PM		
						IM: 0.03 < DEM/DOR < 0.3	CYP2D6 metabolic function is reduced by LK-1 antibody presence		
						PM: DEM/DOR > 0.3			
	Immunosuppressants (tacrolimus or cyclosporine A)	CYP3A5	Reduced activity	1	*CYP3A5*1/*3*	4β-hydroxycholesterol via GCMS	10 *CYP3A5*1* recipients had low CYP3A5 activity (phenoconversion), might be due to increased indoxyl sulfate concentrations	63	[42]
								(kidney transplant recipients)	
Pregnancy	Dextromethorphan	CYP2D6	Reduced activity	2	*CYP2D6*1*/**3*/**4*	Dextromethorphan (30 mg)	NM: metabolic ratio dextromethorphan/dextrorphan ↓ (29%)	140	[43]
						CYP2D6: O-demethylated → dextrorphan (dextromethorphan/dextrorphan)	IM: metabolic ratio dextromethorphan/dextrorphan ↓ (63%)	(pregnant women)	
						CYP3A: *N*-demethylated to 3-methoxymorphinan (dextromethorphan/3-demethoxymorphinon)			
	Paroxetine	CYP2D6	Increased activity	2	*CYP2D6*1*/**3*/**4*/**5*/**6*/**9*/**10*/**41* and duplication	Paroxetine via HPLC-UV or LC-MS/MS	NM/UM: plasma paroxetine concentration ↓	74	[44]
					NM: **1/*1*		PM plasma paroxetine concentration ↑	(pregnant women)	
					UM: duplication				
					IM: heterozygous for nonfunctional allele				
					PM: homozygous for nonfunctional allele				

Type of study refers to “designed with the objective to study phenoconversion” (Type 1) or “not specifically designed to study phenoconversion, did not categorize patients into PM, IM, NM, RM or UM phenotype” (Type 2). HPLC–MS/MS = high pressure liquid chromatography tandem mass spectrometry, GCMS = gas chromatography mass spectrometry, ELISA = enzyme-linked immuno sorbent assay, UM = ultra-rapid metabolizer, NM = normal metabolizer, IM = intermediate metabolizer, PM = poor metabolizer, “→” = converted to, “↑” = increase, “↑↑” = larger increase, “↓” = decrease, “↓↓” = larger decrease.

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
