# Peer review of "Phenoconversion of Cytochrome P450 Metabolism: A Systematic Review"

_jcm, 2020, doi:10.3390/jcm9092890_

Round 1

Reviewer 1 Report

The scientific content of the article is acceptable. However, the reviewer still has concerns about the English grammar in this manuscript and many sentences remain very awkward. Please see my recommendations for improving the English in this manuscript. It is not the job of a reviewer to provide editing services but given that there remained many problems after revision, this reviewer felt obliged. Prior to resubmitting, please insure that this article has been reviewed by a native English speaker as my suggested edits/corrections are not exhaustive.

Lines 40-42: “Most DPWG and CPIC guidelines consider genes encoding cytochrome P450 (CYP450) enzymes involved in human drug metabolism including CYP1A2, CYP2D6, CYP2C9, CYP2C19, CYP3A4 and CYP3A5 that account for the metabolism of 70-80% of approved drugs.”

What do DPWG and CPIC consider about genes encoding CYP450s? Do they consider them clinically actionable or very important pharmacogenes? Editing this sentence to say, “Most DPWG and CPIC guidelines consider genes encoding cytochrome P450 (CYP45) enzymes involved in human drug metabolism, including CYP1A2, CYP2D6, CYP2C9, CYP2C19, CYP3A4 and CYP3A5 that account for the metabolism of 70-80% of approved drugs [5], to be very important pharmacogenes.

Lines 59-60 : Revise the sentence to say, “The effect of phenoconversion may differ by genotype”

Line 79: Replace, “Following” with “Subsequently”

Lines 80-85: Box 1 is still too verbose and repetitive. Consider revising to, “John’s notritypline plasma concentrations resembled those found in a CYP2D6 PM phenotype. If John has been genotyped as CYP2D6 *1/*1, which is interpreted as a CYP2D6 NM phenotype, the drug interaction with amiodraon would have had a minimal or no effect on John’s CYP2D6 phenotype and he would have remained a CYP2D6 NM or shifted to a CYP2D6 IM.”

Lines 89-95 : Consider revising to, “She was genotyped as CYP3A5 *1/*3, which is interpreted as a CYP3A5 IM phenotype [17]. Suzana has been successfully treated with 5 mg tacrolimus b.i.d. since her kidney transplant. However, she recently acquired an infection which caused the related of pro-inflammatory cytokines and increased levels of c-reactive protein (CPR) to 200 mg/L. Suzan has begun to experience tacrolimus-related size effects such as tremor, vomiting and increased serum creatinine concentrations. Suzana’s tacrolimus plasma concentrations were found to be above the therapeutic range which indicates the occurrence of photoconversion from a CYP3A5 IM to PM.

Line 98: A comma is needed between ‘review’ and ‘identification.’

Lines 158-159: Consider revising to ,”The impact of phenoconversion is hypothesized to differ among genotpyes.”

Lines 162-164: Consider revising to ,”For the structure of this review, phenocovnersion is dived into two topics: assessing the effect of 1) concomitant drugs and other extrinsic factors such as smoking and 2) patient- and disease-related factors such as comorbidity and age”

Line 165: A comma is needed between ‘example’ and ‘if.’ Additionally, ‘EM’ was already spelled out above and so it is fine to just use ‘EM.’

Line 211: Please correct the tense to “did no change.”

Line 252: Please correct to “smoking and vitamin.”

Line 285: Please revise to “One paper evaluating phenoconversion related to comedication in patients taking anticoagulant drugs was identified.”

Line 292: The abbreviation and the full spelling of INR should be switched in order.

Lines 294-311 :Please consider revising to revise to: “The study found that during the 2-month follow-up, 28% of patients treated with the CYP2C19 inhibitor, lansoprazole, experienced bleeding events post-surgery compared to patients treated with rabeprazole (0%), which does not affect CYP2C19. Further examination of the outcomes in lansoprazole-treated patients based on patient predicted phenotype revelated that 40% of the CYP2C19 IM patients experienced a bleeding event. Comparatively, only 12% and 22% of CYP2C19 NM and PM patients, respectively, experienced bleeding events thus indicating that CYP2C19 inhibition by lansoprazole has a phenotype-mediated effect in patients treated with warfarin.”

Lines 313-.14: Please revise to, “One paper evaluating co-medication-related phenoconversion in patients taking antihypertensive drugs was included.”

Line 319: Please insert ‘to’ between ‘returned’ and ‘CYP2D6.’

Line 331: ‘With’ should be replayed with ‘by.’

Line 335-336: Please revise to, “For antipsychotics, four papers evaluating phenoconversion were included; three attributed phenoconversion to co-medication and one investigated smoking-related phenocovnersion.”

Line 353: It does makes sense to start a new paragraph here since a new paper with a different mechanism of phenoconversion is being discussed. Please consider putting the new paragraph back.

Line 355: Please insert ‘expression of’ between ‘induces’ and ‘the.’

Line 369: This is an incomplete sentence. Please revise to a complete sentence, perhaps, “The increase in the metabolite to parent ratio indicates reduced CYP1A2 enzyme activity in CYP1A2*1F smokers.

Line 370: Replace ‘by’ with ‘due to.’

Lines 378-379: Please consider revising to, “The authors reported the difference in tramadol and dextromethorphan plasma concentrations between homozygous (CYP2D6 *1/*1) and heterozygous (CYP2d *1/*17) NM subjects,’ for clarity.

Lines 383-387: Please consider revising to, “Nearly all of the heterozygous CYP2D6 NM subjects where phenoconverted with concomitant treatment of paroxetine while 71% of subjects were phenoconverted with duloxetine. Interestingly, in homozygous CYP2D6 IM subjects, 56% were phenoconverted when treated with paroxetine but only 25% were phenoconverted when treated with duloxetine. These results show that the strength of the inhibitor (weak/moderate/strong) is important in the influence of phenoconversion along with genotype.

Line 389: The word ‘used’ can be deleted from the sentence.

Line 397: Please consider revising the sentence to, “One paper examining phenconversion in patients taking cardiac drugs was included.”

Line 399: Inhibitor is incorrectly spelled.

Lines 405-414: Please double check this section to make sure it accurately reflects the study and that increasing and decreasing AUCs are appropriately assigned. Was the metabolic ratio used or just the parent drug? Additionally, this section is a bit verbose and could be rewritten in a more concise manner.

Line 416: Please consider revising sentence to, “A study investigating phenoconversion in patients treated with proton pump inhibitors was included in this review.”

Lines 416-424. These sentences could be condensed to, “In a prospective study of 18 healthy CYP2C19 genotyped volunteers, the effect of fluvoxamine (inhibitor of CYP1A2, CYP2C19 and CYP3A4) on the pharmacokinetics of lansoprazole (CYP2C19 substrate) was investigated. Subjects received lansoprazole with and without concomitant treatment with fluvoxamine [32]. With concomitant fluvoxamine treatment, CYP2C19 inhibition was reported to be genotype dependent with a 2.2- and 1.9-fold decrease for CYP2C19 NM and IM, respectively, compared to administration of lansoprazole alone. This suggests the possible phenoconversion to a lower CYP2C19 metabolizer phenotype (i.e. CYP2C19 IM conversion to IM or PM and IM conversion to PM).

Line 438: Remove ‘did show’ and replace with ‘showed.’

Lines 439-443: These sentences are grammatically incorrect. Consider, “Concomitant treatment with a weak CYP2D6 inhibitor resulted in patients with the CYP2D6 NM genotype to be converted to the IM metabolizer phenotype and IM patients were converted to the PM metabolizer phenotype. This, again, shows that in addition to the genotype, the potency of the inhibitor also plays an important role in the occurrence of phenoconversion.”

Lines 470-471- This sentence does not make sense and is incomplete. Please revise.

Line 496- Remove ‘after’ from sentence.

Lines 500-503: Please consider revising to, “In a prospective study of 52 patients with Behcet disease and 96 health volunteers, the fluence of Behcet disease and CYP2C9 genotype on the activity of CYP2C9 (phenotypical determined by measuring the metabolic ratio of losartan) was determine [39].”

Lines 506-509 : Please consider revising to, “In a prospective study of 31 patients with hepatitis C virus (HCV)- positive chronic hepatitis or cirrhosis and 30 healthy volunteers, the interaction between chronic liver disease and CYP2C19 genotype was assessed [40].”

Line 515: Add ‘respectively’ between ‘(1.9 fold change)’ and ‘which.’

Line 515-18: Either remove ‘trigger phenoconversion’ or cause phenoconversion’ from the sentence. Having both is problematic.

Line 525: ‘Also’ should be removed from the sentence.

Line 533: ‘Reduced’ should be changed to ‘reduction in.’

Lines 534-535: Revise sentence to, 'The metabolizer phenotype was explained by genotype in 3 out of the 10 LKM-1-positive subjects.’

Lines 563-564: Please revise to, “A potential explanation could be that the drugs studies are not specific probes for CYP2D6 and are metabolized by additional CYP450 enzymes that have reduced activity during pregnancy.”

Lines 564-570: This sentence doesn’t make sense. Please revise and clarify.

Author Response

The scientific content of the article is acceptable. However, the reviewer still has concerns about the English grammar in this manuscript and many sentences remain very awkward. Please see my recommendations for improving the English in this manuscript. It is not the job of a reviewer to provide editing services but given that there remained many problems after revision, this reviewer felt obliged. Prior to resubmitting, please insure that this article has been reviewed by a native English speaker as my suggested edits/corrections are not exhaustive. • Response: We thank the reviewer the feedback and all the corrections. We do agree that they reads better, so we replaced the sentences by the advised phrases. Furthermore, we edited other parts of the manuscript on grammar as well. Lines 40-42: “Most DPWG and CPIC guidelines consider genes encoding cytochrome P450 (CYP450) enzymes involved in human drug metabolism including CYP1A2, CYP2D6, CYP2C9, CYP2C19, CYP3A4 and CYP3A5 that account for the metabolism of 70-80% of approved drugs.” What do DPWG and CPIC consider about genes encoding CYP450s? Do they consider them clinically actionable or very important pharmacogenes? Editing this sentence to say, “Most DPWG and CPIC guidelines consider genes encoding cytochrome P450 (CYP45) enzymes involved in human drug metabolism, including CYP1A2, CYP2D6, CYP2C9, CYP2C19, CYP3A4 and CYP3A5 that account for the metabolism of 70-80% of approved drugs [5], to be very important pharmacogenes.
• Original manuscript:
o (lines 40-42) Most DPWG and CPIC guidelines consider genes encoding cytochrome P450 (CYP450) enzymes involved in human drug metabolism including CYP1A2, CYP2D6, CYP2C9, CYP2C19, CYP3A4 and CYP3A5 that account for the metabolism of 70-80% of approved drugs [5].
• Revised manuscript: o (lines 40-43) DPWG and CPIC provide guidelines for genes encoding cytochrome P450 (CYP450) enzymes involved in human drug metabolism including CYP1A2, CYP2D6, CYP2C9, CYP2C19, CYP3A4 and CYP3A5, which account for the metabolism of 70-80% of approved drugs [5]. Lines 59-60 : Revise the sentence to say, “The effect of phenoconversion may differ by genotype”
• Original manuscript:
o (lines 59-60) The effect of phenoconversion may be different per genotype.
• Revised manuscript: o (lines 52-53) The effect of phenoconversion may differ per genotype. Line 79: Replace, “Following” with “Subsequently”
• Original manuscript:
o (lines 79-80) Following, John experienced nortriptyline related side effects, such as tremor, hyperthermia and tachycardia.
• Revised manuscript:
o (lines 72-73) Subsequently, John experienced nortriptyline related side effects, such as tremor, hyperthermia and tachycardia. Lines 80-85: Box 1 is still too verbose and repetitive. Consider revising to, “John’s notritypline plasma concentrations resembled those found in a CYP2D6 PM phenotype. If John has been genotyped as CYP2D6 *1/*1, which is interpreted as a CYP2D6 NM phenotype, the drug interaction with amiodraon
would have had a minimal or no effect on John’s CYP2D6 phenotype and he would have remained a CYP2D6 NM or shifted to a CYP2D6 IM.”
• Original manuscript:
o (lines 80-85) Plasma concentrations showed levels 80 comparable with those found in CYP2D6 PM. Indeed, due to CYP2D6 inhibition by amiodaron John 81 had a CYP2D6 activity concordant with CYP2D6 PM. When John’s CYP2D6 genotype would have 82 been CYP2D6 *1/*1 translating into a CYP2D6 NM phenotype, the drug interaction with amiodaron 83 would have had no or a minimal effect: the CYP2D6 NM phenotype would remain the same or shift 84 to CYP2D6 IM at most.
• Revised manuscript: o (lines 73-78) John’s nortriptyline plasma concentrations resembled those found in a CYP2D6 PM phenotype. If John has been genotyped as CYP2D6 *1/*1, which is interpreted as a CYP2D6 NM phenotype, the drug interaction with amiodarone would have had a minimal or no effect on John’s CYP2D6 phenotype and he would have remained a CYP2D6 NM or shifted to a CYP2D6 IM. Lines 89-95 : Consider revising to, “She was genotyped as CYP3A5 *1/*3, which is interpreted as a CYP3A5 IM phenotype [17]. Suzana has been successfully treated with 5 mg tacrolimus b.i.d. since her kidney transplant. However, she recently acquired an infection which caused the related of pro-inflammatory cytokines and increased levels of c-reactive protein (CPR) to 200 mg/L. Suzan has begun to experience tacrolimus-related size effects such as tremor, vomiting and increased serum creatinine concentrations. Suzana’s tacrolimus plasma concentrations were found to be above the therapeutic range which indicates the occurrence of photoconversion from a CYP3A5 IM to PM.
• Original manuscript:
o (lines 89-95) She was genotyped for CYP3A5 and she is a CYP3A5*1/*3 which translates into a CYP3A5 intermediate metabolizer (IM) phenotype [17]. Suzan has been treated with 5 mg tacrolimus b.i.d. However, recently she acquired an infection which caused the release of pro-inflammatory cytokines and increased levels of c-reactive protein (CRP) (200 mg/L). Following the infection, Suzan experienced tacrolimus related side effects such as tremor, vomiting and increased serum creatinine levels. Measurement of tacrolimus plasma concentrations showed levels above the therapeutic range, indicating the occurrence of phenoconversion from CYP3A5 IM to PM.
• Revised manuscript: o (lines 81-87) She was genotyped as CYP3A5 *1/*3, which is interpreted as a CYP3A5 IM phenotype [17]. Suzan has been successfully treated with 5 mg tacrolimus b.i.d. since her kidney transplant. However, she recently acquired an infection which caused the release of pro-inflammatory cytokines and increased levels of c-reactive protein (CPR) to 200 mg/L. Suzan has begun to experience tacrolimus-related size effects such as tremor, vomiting and increased serum creatinine concentrations. Suzan’s tacrolimus plasma concentrations were found to be above the therapeutic range which indicates the occurrence of phenoconversion from a CYP3A5 IM to PM. Line 98: A comma is needed between ‘review’ and ‘identification.’
• Original manuscript:
o (lines 98-99) For this systematic review identification and selection of studies were performed according to the PICO method [18].
• Revised manuscript: o (lines 90-91) For this systematic review, identification and selection of studies were performed according to the PICO method [18].
Lines 158-159: Consider revising to ,”The impact of phenoconversion is hypothesized to differ among genotpyes.”
• Original manuscript:
o (lines 158-159) The impact of phenoconversion is hypothesized to be different amongst different genotypes.
• Revised manuscript: o (lines 106-107) The impact of phenoconversion is hypothesized to differ amongst different genotypes. Lines 162-164: Consider revising to ,”For the structure of this review, phenocovnersion is dived into two topics: assessing the effect of 1) concomitant drugs and other extrinsic factors such as smoking and 2) patient- and disease-related factors such as comorbidity and age”
• Original manuscript:
o (lines 162-164) For the structure of this review phenoconversion is divided into two topics, assessing the effect of 1) concomitant drugs and other extrinsic factors, such as smoking, and 2) patient- and disease-related factors, such as comorbidity and age.
• Revised manuscript: o (lines 110-112) For the structure of this review, phenoconversion is dived into two topics: assessing the effect of 1) concomitant drugs and other extrinsic factors such as smoking and 2) patient- and disease-related factors such as comorbidity and age. Line 165: A comma is needed between ‘example’ and ‘if.’ Additionally, ‘EM’ was already spelled out above and so it is fine to just use ‘EM.’
• Response: We thank the reviewer for the grammatical correction, we added the comma. Further, this is the first time in the manuscript we use extensive metabolizer, therefore we have the phrase spelled out.
• Original manuscript:
o (lines 165-166) For example if the original paper used the term extensive metabolizers (EM), subjects were assigned NM.
• Revised manuscript: o (lines 113-115) For example, if the original paper used the term extensive metabolizers (EM), subjects were assigned NM. Line 211: Please correct the tense to “did no change.”
• Original manuscript:
o (lines 211) do not change
• Revised manuscript: o (lines 145) did not change Line 252: Please correct to “smoking and vitamin.”
• Original manuscript:
o (lines 252-255) The use of concomitant medication and other extrinsic factors such as alcohol consumption, smoking, vitamin D exposure have all been suggested to influence the occurrence of phenoconversion of CYP450 metabolism and will be described in the next section.
• Revised manuscript:
o (lines 158-161) The use of concomitant medication and other extrinsic factors such as alcohol consumption, smoking and vitamin D exposure, have all been suggested to influence the occurrence of phenoconversion of CYP450 metabolism, and will be described in the next section. Line 285: Please revise to “One paper evaluating phenoconversion related to comedication in patients taking anticoagulant drugs was identified.”
• Original manuscript:
o (lines 286-287) For anticoagulant drugs one paper evaluating phenoconversion was identified related to co-medication-induced phenoconversion.
• Revised manuscript: o (lines 192-193) One paper evaluating phenoconversion related to comedication in patients taking anticoagulant drugs was identified. Line 292: The abbreviation and the full spelling of INR should be switched in order.
• Original manuscript:
o (lines 291-292) Warfarin dose prior to concomitant treatment with a proton-pump inhibitor was based on their INR (international normalized ratio).
• Revised manuscript: o (lines 198-199) Warfarin dose prior to concomitant treatment with a proton-pump inhibitor was based on their international normalized ratio (INR). Lines 294-311 :Please consider revising to revise to: “The study found that during the 2-month follow-up, 28% of patients treated with the CYP2C19 inhibitor, lansoprazole, experienced bleeding events post-surgery compared to patients treated with rabeprazole (0%), which does not affect CYP2C19. Further examination of the outcomes in lansoprazole-treated patients based on patient predicted phenotype revelated that 40% of the CYP2C19 IM patients experienced a bleeding event. Comparatively, only 12% and 22% of CYP2C19 NM and PM patients, respectively, experienced bleeding events thus indicating that CYP2C19 inhibition by lansoprazole has a phenotype-mediated effect in patients treated with warfarin.”
• Original manuscript:
o (lines 294-311) This study showed during the 2-month follow-up that patients treated with the CYP2C19 inhibitor lansoprazole (24%) had an increased risk of developing bleeding events post-surgery compared to users of the rabeprazole (0%) that did not affect CYP2C19. Further examining this outcome for the different predicted phenotypes revealed that 40% of the CYP2C19 IM patients who concomitantly received lansoprazole developed a bleeding event, which was greater than the incidences of bleedings seen in the CYP2C19 NM (12%) and PM (22%) patients. Indicating, the distinct effect of the CYP2C19 inhibitor lansoprazole on the different CYP2C19 metabolizer phenotype groups treated with warfarin.
• Revised manuscript: o (lines 203-209) The study found that during the 2-month follow-up, 24% of patients treated with the CYP2C19 inhibitor, lansoprazole, experienced bleeding events post-surgery compared to patients treated with rabeprazole (0%), which does not affect CYP2C19. Further examination of the outcomes in lansoprazole-treated patients based on patient predicted phenotype revelated that 40% of the CYP2C19 IM patients experienced a bleeding event. Comparatively, only 12% and 22% of CYP2C19 NM and PM patients, respectively, experienced bleeding events thus indicating that CYP2C19 inhibition by lansoprazole has a phenotype-mediated effect in patients treated with warfarin.
Lines 313-.14: Please revise to, “One paper evaluating co-medication-related phenoconversion in patients taking antihypertensive drugs was included.”
• Original manuscript:
o (lines 313-314) For antihypertensive drugs one paper evaluating co-medication-related phenoconversion was included.
• Revised manuscript: o (lines 219-220) One paper evaluating co-medication-related phenoconversion in patients taking antihypertensive drugs was included. Line 319: Please insert ‘to’ between ‘returned’ and ‘CYP2D6.’
• Original manuscript:
o (lines 319-320) All of the CYP2D6 NM genotype patients returned CYP2D6 phenotypical NMs after complete withdrawal.
• Revised manuscript: o (lines 225-226) All of the CYP2D6 NM genotype patients returned to CYP2D6 phenotypical NMs after complete withdrawal. Line 331: ‘With’ should be replayed with ‘by.’
• Original manuscript:
o (lines 330-331) The authors reported that the oral clearance of tolterodine decreased by 80% in CYP2D6 NM subjects and with 93% in CYP2D6 IM subjects.
• Revised manuscript: o (lines 237-239) The authors reported that the oral clearance of tolterodine decreased by 80% in CYP2D6 NM subjects and by 93% in CYP2D6 IM subjects Line 335-336: Please revise to, “For antipsychotics, four papers evaluating phenoconversion were included; three attributed phenoconversion to co-medication and one investigated smoking-related phenocovnersion.”
• Original manuscript:
o (lines 335-336) For antipsychotics three four papers evaluating phenoconversion were included. Three attributed to co-medication-related phenoconversion and one investigating smoking-related phenoconversion.
• Revised manuscript: o (lines 243-244) For antipsychotics, four papers evaluating phenoconversion were included; three attributed phenoconversion to co-medication and one investigated smoking-related phenoconversion. Line 353: It does makes sense to start a new paragraph here since a new paper with a different mechanism of phenoconversion is being discussed. Please consider putting the new paragraph back. o Response: We thank the reviewer for the comment, we agree that it is a different mechanism so a new paragraph is started Line 355: Please insert ‘expression of’ between ‘induces’ and ‘the.’
• Original manuscript:
o (lines 354-355) Once activated, the aryl hydrocarbon receptor induces the CYP1A1 and CYP1A2 enzymes [49].
• Revised manuscript:
o (lines 263-264) Once activated, the aryl hydrocarbon receptor induces expression of the CYP1A1 and CYP1A2 enzymes [49]. Line 369: This is an incomplete sentence. Please revise to a complete sentence, perhaps, “The increase in the metabolite to parent ratio indicates reduced CYP1A2 enzyme activity in CYP1A2*1F smokers.
• Original manuscript:
o (lines 369) Indicating reduced CYP1A2 enzyme activity in CYP1A2*1F smokers.
• Revised manuscript: o (lines 273-274) The increase in the metabolite to parent ratio indicates reduced CYP1A2 enzyme activity in CYP1A2*1F smokers. Line 370: Replace ‘by’ with ‘due to.’
• Original manuscript:
o (lines 369-371) This study suggests that phenoconversion by smoking is strongly influenced by the CYP1A2 genotype with as consequence that CYP1A2*1F smokers may require higher doses of olanzapine.
• Revised manuscript: o (lines 274-275) This study suggests that phenoconversion due to smoking is strongly influenced by the CYP1A2 genotype with as consequence that CYP1A2*1F smokers may require higher doses of olanzapine. Lines 378-379: Please consider revising to, “The authors reported the difference in tramadol and dextromethorphan plasma concentrations between homozygous (CYP2D6 *1/*1) and heterozygous (CYP2d *1/*17) NM subjects,’ for clarity.
• Original manuscript:
o (lines 378-379) The authors reported the difference between homozygous (CYP2D6 *1/*1) and heterozygous (CYP2D6 *1/*17) NM subjects.
• Revised manuscript: o (lines 283-285) The authors reported the difference in tramadol and dextromethorphan plasma concentrations between homozygous (CYP2D6 *1/*1) and heterozygous (CYP2d *1/*17) NM subjects. Lines 383-387: Please consider revising to, “Nearly all of the heterozygous CYP2D6 NM subjects where phenoconverted with concomitant treatment of paroxetine while 71% of subjects were phenoconverted with duloxetine. Interestingly, in homozygous CYP2D6 IM subjects, 56% were phenoconverted when treated with paroxetine but only 25% were phenoconverted when treated with duloxetine. These results show that the strength of the inhibitor (weak/moderate/strong) is important in the influence of phenoconversion along with genotype.
• Original manuscript:
o (lines 383-387) Almost all of the heterozygous CYP2D6 NM subjects were phenoconverted due to the strong inhibitor paroxetine. For the moderate inhibitor duloxetine this was only 71%. This difference was even bigger in homozygous CYP2D6 IM subjects, 56% vs 25% for paroxetine and duloxetine, respectively. These results show that the strength of the inhibitor (weak/moderate/strong) important also influences the occurrence of phenoconversion.
• Revised manuscript: o (lines 290-295) Nearly all of the heterozygous CYP2D6 NM subjects where phenoconverted with concomitant treatment of paroxetine while 71% of subjects were phenoconverted with duloxetine. Interestingly, in homozygous CYP2D6 IM subjects,
56% were phenoconverted when treated with paroxetine but only 25% were phenoconverted when treated with duloxetine. These results show that the strength of the inhibitor (weak/moderate/strong) is important in the influence of phenoconversion along with genotype. Line 389: The word ‘used’ can be deleted from the sentence.
• Original manuscript:
o (lines 387-390) In a follow-up in vitro study using human liver microsomes, the authors showed that the phenoconversion is likely to be explained through a difference in amount of functional CYP2D6 enzyme and not by differences in the potency of the used CYP2D6 inhibitors for the different genotypes [50].
• Revised manuscript: o (lines 295-298) In a follow-up in vitro study using human liver microsomes, the authors showed that the phenoconversion is likely to be explained through a difference in amount of functional CYP2D6 enzyme and not by differences in the potency of the CYP2D6 inhibitors for the different genotypes [50]. Line 397: Please consider revising the sentence to, “One paper examining phenconversion in patients taking cardiac drugs was included.”
• Original manuscript:
o (lines 397) There was one paper showing phenoconversion for cardiac drugs.
• Revised manuscript: o (lines 304) One paper examining phenoconversion in patients taking cardiac drugs was included. Line 399: Inhibitor is incorrectly spelled.
• Original manuscript:
o (lines 397-399) In a prospective study 143 supraventricular tachyarrhythmia patients were concomitantly treated with flecainide and bepridil, a CYP2D6 inhibiter [51].
• Revised manuscript: o (lines 305-307) In a prospective study 143 supraventricular tachyarrhythmia patients were concomitantly treated with flecainide and bepridil, a CYP2D6 inhibitor [51]. Lines 405-414: Please double check this section to make sure it accurately reflects the study and that increasing and decreasing AUCs are appropriately assigned. Was the metabolic ratio used or just the parent drug? Additionally, this section is a bit verbose and could be rewritten in a more concise manner.
• Original manuscript:
o (lines 405-414) One study examined phenoconversion by concomitant medication in patients using NSAIDs. This prospective cross-over study examined in 22 healthy volunteers the consequences of 7-days concomitant treatment with the CYP2C9 inhibitor fluconazole for the metabolism of the CYP2C9 substrate flurbiprofen [31]. Co-administration of 200 mg fluconazole increased the AUC of flurbiprofen in the 11 CYP2C9 predicted NM to an extent that was typical for CYP2C9 IM. Higher dosages of fluconazole (400 mg) changed the AUC even further to an extent that was indicative for the conversion into CYP2C9 PM. On the other hand, both dosages of fluconazole reduced the AUC of flurbiprofen in the 8 predicted IM subjects to levels that were representative for CYP2C9 PM subjects. These results indicate that concomitant use of fluconazole can cause phenoconversion in a dose- and genotype-dependent manner.
• Revised manuscript: o (lines 313-327) One study examined phenoconversion by concomitant medication in patients using NSAIDs. In a prospective cross-over study of 22 healthy volunteers, the consequences of 7-days concomitant treatment with flurbiprofen (a CYP2C9 substrate) and fluconazole (a CYP2C9 inhibitor) was investigated [31]. Co-administration of 200 mg fluconazole increased the AUC of flurbiprofen in predicted CYP2C9 NM (n=11) to a AUC which is comparable with the AUC of CYP2C9 IM subjects (n=8), and a dose of 400 mg fluconazole increased the AUC even further comparable to CYP2C9 PMs (n=2). In predicted CYP2C9 IM subjects, both dosages of fluconazole increased the AUC of flurbiprofen comparable with the AUCs of CYP2C9 PM subjects. These results indicate that concomitant use of fluconazole can cause phenoconversion in a dose- and genotype-dependent manner. Line 416: Please consider revising sentence to, “A study investigating phenoconversion in patients treated with proton pump inhibitors was included in this review.”
• Original manuscript:
o (lines 416) There was one paper showing phenoconversion for proton pump inhibitors.
• Revised manuscript: o (lines 331-332) A study investigating phenoconversion in patients treated with proton pump inhibitors was included in this review. Lines 416-424. These sentences could be condensed to, “In a prospective study of 18 healthy CYP2C19 genotyped volunteers, the effect of fluvoxamine (inhibitor of CYP1A2, CYP2C19 and CYP3A4) on the pharmacokinetics of lansoprazole (CYP2C19 substrate) was investigated. Subjects received lansoprazole with and without concomitant treatment with fluvoxamine [32]. With concomitant fluvoxamine treatment, CYP2C19 inhibition was reported to be genotype dependent with a 2.2- and 1.9-fold decrease for CYP2C19 NM and IM, respectively, compared to administration of lansoprazole alone. This suggests the possible phenoconversion to a lower CYP2C19 metabolizer phenotype (i.e. CYP2C19 IM conversion to IM or PM and IM conversion to PM).
• Original manuscript:
o (lines 416-424) In a prospective study of 18 healthy volunteers the effect of fluvoxamine on the pharmacokinetics of lansoprazole among different CYP2C19 genotypes was investigated. Subjects received the CYP2C19 substrate lansoprazole with or without concomitant treatment with fluvoxamine [32]. During concomitant treatment the level of CYP2C19 inhibition was reported to be genotype dependent with a 2.2 fold decrease for CYP2C19 NMs and 1.9 fold for CYP2C19 IMs, suggesting the possible conversion to a lower CYP2C19 metabolizer phenotype (i.e. CYP2C19 NM conversion into the CYP2C19 IM or CYP2C19 PM metabolizer phenotype, and CYP2C19 IM conversion into the CYP2C19 PM metabolizer phenotype).
• Revised manuscript:
o (lines 333-340) In a prospective study of 18 healthy CYP2C19 genotyped volunteers, the effect of fluvoxamine (inhibitor of CYP1A2, CYP2C19 and CYP3A4) on the pharmacokinetics of lansoprazole (CYP2C19 substrate) was investigated. Subjects received lansoprazole with and without concomitant treatment with fluvoxamine [32]. With concomitant fluvoxamine treatment, CYP2C19 inhibition was reported to be genotype dependent with a 2.2- and 1.9-fold decrease for CYP2C19 NM and IM, respectively, compared to administration of lansoprazole alone. This suggests the possible phenoconversion to a lower CYP2C19 metabolizer phenotype (i.e. CYP2C19 IM conversion to IM or PM and IM conversion to PM). Line 438: Remove ‘did show’ and replace with ‘showed.’
• Original manuscript:
o (lines 437-439) The CYP2D6 UM patients treated with weak or potent CYP2D6 inhibitors did show lower endoxifen plasma concentrations, comparable to the CYP2D6 NM or IM metabolizer phenotype.
• Revised manuscript: o (lines 360-362) The CYP2D6 UM patients treated with weak or potent CYP2D6 inhibitors showed lower endoxifen plasma concentrations, comparable to the CYP2D6 NM or IM metabolizer phenotype. Lines 439-443: These sentences are grammatically incorrect. Consider, “Concomitant treatment with a weak CYP2D6 inhibitor resulted in patients with the CYP2D6 NM genotype to be converted to the IM metabolizer phenotype and IM patients were converted to the PM metabolizer phenotype. This, again, shows that in addition to the genotype, the potency of the inhibitor also plays an important role in the occurrence of phenoconversion.”
• Original manuscript:
o (lines 439-443) Concomitant treatment with a weak inhibitor CYP2D6 inhibitor resulted for patients with the CYP2D6 NM genotype in conversion to the CYP2D6 IM metabolizer phenotype and CYP2D6 IM patients were converted to the CYP2D6 PM metabolizer phenotype. This, again, shows that in addition to the genotype also the potency of the inhibitor plays an important role in the occurrence of phenoconversion.
• Revised manuscript: o (lines 362-366) Concomitant treatment with a weak CYP2D6 inhibitor resulted in patients with the CYP2D6 NM genotype to be converted to the IM metabolizer phenotype and IM patients were converted to the PM metabolizer phenotype. This, again, shows that in addition to the genotype, the potency of the inhibitor also plays an important role in the occurrence of phenoconversion. Lines 470-471- This sentence does not make sense and is incomplete. Please revise.
• Original manuscript:
o (lines 470-471) For the younger subjects there were no phenotypically CYP2C19 PM subject of the CYP2C19 NM or IM genotype groups.
• Revised manuscript: o (lines 395-396) There were no phenotypical CYP2C19 PM subjects in the genotypical CYP2C19 NM or IM groups among the younger subjects. Line 496- Remove ‘after’ from sentence.
• Original manuscript:
o (lines 496-498) However, after based on proguanil metabolism, 27% of the genotypically predicted CYP2C19 NMs and 53% of the genotypically predicted CYP2C19 IMs were phenotypically CYP2C19 PMs.
• Revised manuscript: o (lines 422-424) However, based on proguanil metabolism, 27% of the genotypically predicted CYP2C19 NMs and 53% of the genotypically predicted CYP2C19 IMs were phenotypically CYP2C19 PMs. Lines 500-503: Please consider revising to, “In a prospective study of 52 patients with Behcet disease and 96 health volunteers, the fluence of Behcet disease and CYP2C9 genotype on the activity of CYP2C9 (phenotypical determined by measuring the metabolic ratio of losartan) was determine [39].”
• Original manuscript:
o (lines 500-503) In a prospective study in 52 patients with Behçet disease and 96 healthy volunteers the aim was to investigate the influence of Behçet disease and CYP2C9 genotype on the activity of CYP2C9, which was phenotypically determined by measuring the metabolic ratio of losartan [39].
• Revised manuscript: o (lines 426-429) In a prospective study of 52 patients with Behcet disease and 96 health volunteers, the fluence of Behcet disease and CYP2C9 genotype on the activity of CYP2C9 (phenotypical determined by measuring the metabolic ratio of losartan) was determine [39]. Lines 506-509 : Please consider revising to, “In a prospective study of 31 patients with hepatitis C virus (HCV)- positive chronic hepatitis or cirrhosis and 30 healthy volunteers, the interaction between chronic liver disease and CYP2C19 genotype was assessed [40].”
• Original manuscript:
o (lines 506-509) In a prospective study in 31 patients with hepatitis C virus (HCV)–positive chronic hepatitis or cirrhosis and 30 healthy volunteers the aim was to assess the interaction between chronic liver disease and CYP2C19 genotype [40].
• Revised manuscript: o (lines 435-437) In a prospective study of 31 patients with hepatitis C virus (HCV)- positive chronic hepatitis or cirrhosis and 30 healthy volunteers, the interaction between chronic liver disease and CYP2C19 genotype was assessed [40]. Line 515: Add ‘respectively’ between ‘(1.9 fold change)’ and ‘which.’
• Original manuscript:
o (lines 511-516) In contrast, patients with chronic liver disease genotypically classified as CYP2C19 NM, EM and PM displayed metabolic ratios of 17.15 (21.1 fold change), 20.02 (12.4 fold change) and 26.04 (1.9 fold change), which would phenotypically classify them all as poor metabolizers of CYP2C19.
• Revised manuscript: o (lines 442-445) In contrast, patients with chronic liver disease genotypically classified as CYP2C19 NM, EM and PM displayed metabolic ratios of 17.15 (21.1 fold change), 20.02 (12.4 fold change) and 26.04 (1.9 fold change), respectively, which would phenotypically classify them all as poor metabolizers of CYP2C19. Line 515-18: Either remove ‘trigger phenoconversion’ or cause phenoconversion’ from the sentence. Having both is problematic.
• Original manuscript:
o (lines 515-518) These results demonstrate that chronic liver disease resulting from HCV infection can reduce CYP2C19 enzymatic activity and trigger phenoconversion cause phenoconversion in an infection- and genotype-dependent manner.
• Revised manuscript: o (lines 445-447) These results demonstrate that chronic liver disease resulting from HCV infection can reduce CYP2C19 enzymatic activity and cause phenoconversion in an infection- and genotype-dependent manner. Line 525: ‘Also’ should be removed from the sentence.
• Original manuscript:
o (lines 524-526) In a study using microsomes from 114 organ transplants described above (see alcohol consumption) also the effect of inflammation on phenoconversion was investigated [8].
• Revised manuscript: o (lines 453-455) In a study using microsomes from 114 organ transplants described above (see alcohol consumption) the effect of inflammation on phenoconversion was investigated [8]. Line 533: ‘Reduced’ should be changed to ‘reduction in.’
• Original manuscript:
o (lines 532-534) When LKM-1 negative patients were compared to LKM-1 positive patients, up to six-fold reduced CYP2D6 metabolic activity was found in patients with a high level of LKM-1 antibody (the positive patients).
• Revised manuscript: o (lines 461-463) When LKM-1 negative patients were compared to LKM-1 positive patients, up to six-fold reduction CYP2D6 metabolic activity was found in patients with a high level of LKM-1 antibody (the positive patients). Lines 534-535: Revise sentence to, 'The metabolizer phenotype was explained by genotype in 3 out of the 10 LKM-1-positive subjects.’
• Original manuscript:
o (lines 534-535) For only 3 of the 10 LKM-1 positive subjects their metabolizer phenotype was explained by their genotype.
• Revised manuscript: o (lines 463-464) The metabolizer phenotype was explained by genotype in 3 out of the 10 LKM-1-positive subjects. Lines 563-564: Please revise to, “A potential explanation could be that the drugs studies are not specific probes for CYP2D6 and are metabolized by additional CYP450 enzymes that have reduced activity during pregnancy.”
• Original manuscript:
o (lines 563-564) A potential explanation could be metabolism of the studied drugs by other CYP450 enzymes, which activity is reduced during pregnancy.
• Revised manuscript: o (lines 494-496) A potential explanation could be that the drugs studies are not specific probes for CYP2D6 and are metabolized by additional CYP450 enzymes that have reduced activity during pregnancy. Lines 564-570: This sentence doesn’t make sense. Please revise and clarify.
• Original manuscript:
o (lines 564-570) For CYP2D6 PMs this will result in an altered metabolic capacity, whereby for CYP2D6 IM subjects both processes will middle out [45].
• Revised manuscript: o (lines 497-499) For CYP2D6 PMs the result is an altered metabolic capacity, for CYP2D6 IM subjects both processes (induced and reduced metabolic capacity) will middle out, resulting in an unaffected plasma concentration [

Reviewer 2 Report

The authors have responded to comments satisfactorily.

Author Response

The authors have responded to comments satisfactorily.

• Response: We thank the reviewer for revising our manuscript.

Reviewer 3 Report

The authors have made extensive edits an updates in responding to this reviewers initial comments. This review is a valuable contribution to the literature, being a comprehensive review of papers that will serve a reference source for clinicians and researchers alike.

I have no major comments to add. The addition of the broader phenotype categories put the context of phenoconversion to increased metabolic phenotypes in context. The consideration of different levels of inhibitors was also addressed well. The added cases are illustrative and provide good examples. The comprehensive table is a very good resource. The authors should be commended for their work on this paper.

Minor comments

Italicize genes throughout

Line 79 "amiodaron" should be "amiodarone"

Line 327 Add a space between "Antimuscarinics" and "One"

Author Response

The authors have made extensive edits an updates in responding to this reviewers initial comments. This review is a valuable contribution to the literature, being a comprehensive review of papers that will serve a reference source for clinicians and researchers alike. I have no major comments to add. The addition of the broader phenotype categories put the context of phenoconversion to increased metabolic phenotypes in context. The consideration of different levels of inhibitors was also addressed well. The added cases are illustrative and provide good examples. The comprehensive table is a very good resource. The authors should be commended for their work on this paper. Minor comments Italicize genes throughout
• Response: We thank the reviewer for this comment, we italicized all the genes in the manuscript. Line 79 "amiodaron" should be "amiodarone"
• Response: We thank the reviewer for spotting this typo, we corrected it.
• Original manuscript:
o (lines 78-79) John has recently been diagnosed with arrhythmia and has therefore started therapy with the weak CYP2D6 inhibitor amiodaron [16].
• Revised manuscript: o (lines 71-72) John has recently been diagnosed with arrhythmia and has therefore started therapy with the weak CYP2D6 inhibitor amiodarone [16]. Line 327 Add a space between "Antimuscarinics" and "One"
o Response: We thank the reviewer for spotting this mistake, a enter is added to separate “antimuscarinics” and “One”.

This manuscript is a resubmission of an earlier submission. The following is a list of the peer review reports and author responses from that submission.

Round 1

Reviewer 1 Report

This manuscript is a systematic review the literature to identify extrinsic and intrinsic factors that contribute to phenoconversion. This is an emerging area of research in pharmacogenomics with increasing importance if we are to truly achieve individualized medicine in a clinical setting.

My largest technical concern is that the survey of the literature for study inclusion might not have been as inclusive and extensive as it could have been. I noticed that papers investigating phenoconversion from the Ibero-Latino-American Network of Pharmacogenetics and Pharmacogenomics (RIBEF) were missing from this review. One paper in particular from this group (De Andrés F, et al. OMICS. 2016; DOI: 10.1089/omi.2016.0148) investigated the incidence of phenoconversion in 139 unrelated, medication-free, healthy Ecuadorians using genotyping and a probe cocktail. Interestingly, while the authors reported finding phenotype-genotype discordance (phenoconversion) in the population, they did not find significant influences from smoking, alcohol consumption, or gender. However, they did report that caffeine influenced CYP1A2 phenotype-genotype concordance. Furthermore, while this review cited Shah and Smith’s 2014 BJCP review, “Addressing phenoconversion: the Achilles’ heel of personalized medicine,” I noticed that many of the articles cited in that review that would be relevant to include in this review were not- I think only three of were. I find this odd given that the authors state in Lines 99-100 in the method section that reference lists from reviewers were manually checked to identify relevant cross-references.

My largest technical concern is the English grammar (especially sentence structure) and punctuation in this manuscript. The entire manuscript needs to be edited but section 3.3 was particularly rough.

Below are additional line-by-line comments, questions and suggestions:

Lines 40-42: Zanger’s review actually suggests CYP2D6, CYP2C9 and CYP2C19 account for the metabolism of 39.6% of approved drugs (Figure 1 of his review), not 70-80%. This needs to be corrected.

Line 45: A citation is needed

Line 86: This is the first use of CRP in the manuscript and should be spelled out first here. Instead, it is first spelled out in Line 86.

Both Box 1 and Box 2 are extremely verbose. These should be edited for succinctness and clarity.

Line 121: Be consistent with writing out numbers versus using Arabic numbers for all values greater than 7 (in this case, sixteen is written out instead of using 16).

In Table 1, check your spelling of carbamazepine as it is consistently spelled “carbamazeine” in the tables.

In Table 1, for the mephenytoin study (reference 8), instead of listing “other” as the cause of phenoconversion, why not list it a “dietary”. Additionally, since this particular study is used in Tables 1 and 2, to avoid confusion, it would be better to parse the findings out between the two tables instead of refereeing to all the findings twice.

In Table 1, for the olanzapine study, why not list “smoking” as the cause of phenoconversion and then list “nicotine” instead of “smoking” as the drug responsible for phenoconversion?

Lines 140-145- UMR is never defined here but is used in the table.

In Table 2, is it really appropriate to refer to pregnancy and age as causing inhibition under “Type of interaction”? There is a difference between chemical inhibition of an enzyme and downregulation of enzyme expression. The mechanisms in which intrinsic factors seemingly reduce enzyme activity are generally not elucidated in these studies. Perhaps it would be better to label the column ‘Result’ or something else. That way, the authors could use “reduced activity” and “increased activity” instead of implying a mechanism that has not, in most cases been elucidated.

Subsections would be helpful for sections 3.3 and 3.4 (ex. 3.3.1 Anticonvulsant, 3.4.2 Cancer)

Line 178: Clarify what stiripentol is a strong inhibitor of.

Lines 221-223: This sentence does not make sense. What are the 132% and 107% increase in 4’-N-desmethylolanzapine/olanzapine ratio due to?

Line 306: Please elaborate on what the phenoconversion was associated with or thought to be associated with according to the authors.

Line 378: Again, the mechanisms of reduced enzyme activity due to intrinsic factors have in most cases not been evaluated as with my comment about table 2. Differentiate between chemical inhibition and reduction of enzyme expression. Inhibition of the enzymes seems to be due to inflammation in some cases but what about age and pregnancy? This could be due to the reduction of enzyme expression, which is not chemical inhibition. Be careful to not overly generalize here.

Lines 444-445. This sentence is confusing and doesn’t make much sense. Please revise.

Author Response

This manuscript is a systematic review the literature to identify extrinsic and intrinsic factors that contribute to phenoconversion. This is an emerging area of research in pharmacogenomics with increasing importance if we are to truly achieve individualized medicine in a clinical setting.

My largest technical concern is that the survey of the literature for study inclusion might not have been as inclusive and extensive as it could have been. I noticed that papers investigating phenoconversion from the Ibero-Latino-American Network of Pharmacogenetics and Pharmacogenomics (RIBEF) were missing from this review. One paper in particular from this group (De Andrés F, et al. OMICS. 2016; DOI: 10.1089/omi.2016.0148) investigated the incidence of phenoconversion in 139 unrelated, medication-free, healthy Ecuadorians using genotyping and a probe cocktail. Interestingly, while the authors reported finding phenotype-genotype discordance (phenoconversion) in the population, they did not find significant influences from smoking, alcohol consumption, or gender. However, they did report that caffeine influenced CYP1A2 phenotype-genotype concordance. Furthermore, while this review cited Shah and Smith’s 2014 BJCP review, “Addressing phenoconversion: the Achilles’ heel of personalized medicine,” I noticed that many of the articles cited in that review that would be relevant to include in this review were not- I think only three of were. I find this odd given that the authors state in Lines 99-100 in the method section that reference lists from reviewers were manually checked to identify relevant cross-references.

  • Response: We thank the reviewer for this valuable comment regarding our search strategy. To address this comment, we have revisited our search strategy and carefully looked at the terms and papers that appear in reviews as suggested by the reviewer. As a result, we have expanded the search string with the terms intermediate, rapid and ultra-rapid metabolizer. This has resulted in the inclusion of 2 new studies. Furthermore, reexamination of the studies in the search string by comparing to the review from Shah and Smith 2014 added 2 more studies.

We thank the reviewer for referring us to a potential phenoconversion study. In our definition of phenoconversion, we focused on the influence of non-genetic factors as potential explanation for the observed discrepancies between genotype and metabolizer phenotype. For phenoconversion resulting from the use of concomitant medication we aimed to make a clear distinction between traditional drug-drug interaction studies, in which the genotype is not taken in consideration, and drug-drug-gene interaction studies. The exact criteria are formulated in the methods section (page 3 line 127-135). One important criteria for studies investigating phenoconversion resulting from concomitant medication was that the effect of phenoconversion should be studied for different genotype-based predicted phenotypes, as this would allow conclusions whether and how the outcome of drug-drug interactions would be modulated by the different genotypes.

Due to our definition of phenoconversion, and the formulated inclusion criteria, not all studies described by Shah and Smith 2014 were included.

The study of De Andrés et al. suggested by the reviewer indeed investigates phenoconversion in healthy subjects. However, this study does not investigate the effect on specific genotypical predicted phenotypes, this study therefore did not match the inclusion criteria and was therefore not included in the review.

The following sections were added to the manuscript:

    • Revised manuscript:
      • (lines 222-237 and added in table 1) For anticoagulant drugs one paper evaluating phenoconversion was identified related to co-medication-induced phenoconversion. In a prospective study of 82 heart surgery patients (CYP2C9 *1/*1 and VKORC1 T/T) the influence of the CYP2C19 genotype on the occurrence of bleeding events during treatment with warfarin was investigated [24]. Half of the patients (n=41) were concomitantly treated with lansoprazole (a CYP2C19 inhibitor) and the other half (n=41) with rabeprazole (control group). Warfarin dose prior to concomitant treatment with a proton-pump inhibitor was based on their INR (international normalized ratio). The outcome of the study was the number of bleeding events, such as tarry stool, bleeding from colon diverticulum, conjunctival bleeding, or bleeding into the shoulder joint or surgical sites. This study showed during the 2-month follow-up that patients treated with the CYP2C19 inhibitor lansoprazole (24%) had an increased risk of developing bleeding events post-surgery compared to users of the rabeprazole (0%) that did not affect CYP2C19. Further examining this outcome for the different predicted phenotypes revealed that 40% of the CYP2C19 IM patients who concomitantly received lansoprazole developed a bleeding event, which was greater than the incidences of bleedings seen in the CYP2C19 NM (12%) and PM (22%) patients. Indicating, the distinct effect of the CYP2C19 inhibitor lansoprazole on the different CYP2C19 metabolizer phenotype groups treated with warfarin.
      • (lines 239-252 and added in table 1) For antihypertensive drugs one paper evaluating co-medication-related phenoconversion was included. In a prospective study of 16 male psychiatric patients treated with thioridazine, the influence of thioridazine withdrawal on the debrisoquine (a CYP2D6 probe drug) MR was evaluated together with the effect of CYP2D6 genotype [25]. When treated with thioridazine (150 mg/d or higher) fourteen patients (87.5%) were phenotypical CYP2D6 PM subjects (MR > 12.6). After complete withdrawal of thioridazine in ten patients, only two genotypical CYP2D6 PM patients remained phenotypical CYP2D6 PM. All of the CYP2D6 NM genotype patients returned CYP2D6 phenotypical NMs after complete withdrawal. At 50 and 100 mg of thioridazine the proportion of phenotypical CYP2D6 NM subjects decreased to 33% and 29%, respectively. Genotypic CYP2D6 IM patients receiving 50 mg of thioridazine per day, were all converted to the CYP2D6 PM metabolizer phenotype, compared to 67% of the genotypic CYP2D6 NM patients. These results show that genotypic CYP2D6 IMs are more susceptible to phenoconversion by thioridazine compared to CYP2D6 NM. Moreover, these results indicate that concomitant use of thioridazine results in phenoconversion with a dose dependent effect.
      • (lines 328-337 and added in table 1) One study examined phenoconversion by concomitant medication in patients using NSAIDs. This prospective cross-over study examined in 22 healthy volunteers the consequences of 7-days concomitant treatment with the CYP2C9 inhibitor fluconazole for the metabolism of the CYP2C9 substrate flurbiprofen [31]. Co-administration of 200 mg fluconazole increased the AUC of flurbiprofen in the 11 CYP2C9 predicted NM to an extent that was typical for CYP2C9 IM. Higher dosages of fluconazole (400 mg) changed the AUC even further to an extent that was indicative for the conversion into CYP2C9 PM. On the other hand, both dosages of fluconazole reduced the AUC of flurbiprofen in the 8 predicted IM subjects to levels that were representative for CYP2C9 PM subjects. These results indicate that concomitant use of fluconazole can cause phenoconversion in a dose- and genotype-dependent manner.
      • (lines 351-366 and added in table 1) Two papers investigated phenoconversion in patients using anti-estrogenic drugs, one paper due to concomitant medication and one paper due to vitamin D exposure. In a prospective cohort study 158 breast cancer patients were treated with tamoxifen and received concomitant treatment with CYP2D6 inhibitors [33]. The study aim was to show that concomitant treatment with CYP2D6 inhibitors reduces the endoxifen plasma concentration. Out of the 158 patients, 17 patients used potent CYP2D6 inhibitors (paroxetine and fluoxetine) and 25 patients used weak CYP2D6 inhibitors (sertraline, citalopram, celecoxib, diphenydramine, and chlorpheniramine). Eighty-five patients not taking any concomitant CYP2D6 medication were used as control group. With the exception of CYP2D6 UM, all patients receiving concomitant treatment with a strong CYP2D6 inhibitor were phenoconverted to the CYP2D6 PM metabolizer phenotype. The CYP2D6 UM patients treated with weak or potent CYP2D6 inhibitors did show lower endoxifen plasma concentrations, comparable to the CYP2D6 NM or IM metabolizer phenotype. Concomitant treatment with a weak inhibitor CYP2D6 inhibitor resulted for patients with the CYP2D6 NM genotype in conversion to the CYP2D6 IM metabolizer phenotype and CYP2D6 IM patients were converted to the CYP2D6 PM metabolizer phenotype. This, again, shows that in addition to the genotype also the potency of the inhibitor plays an important role in the occurrence of phenoconversion.
      • (lines 421-427 and added in table 2) In a prospective study in 52 patients with Behçet disease and 96 healthy volunteers the aim was to investigate the influence of Behçet disease and CYP2C9 genotype on the activity of CYP2C9, which was phenotypically determined by measuring the metabolic ratio of losartan [39]. The 31 patients that were genotypically classified as CYP2C9 NM had a mean metabolic ratio that was comparable to the observed metabolic ratio of losartan in CYP2C9 IM healthy volunteers. These results indicate that Behçet disease, an systemic inflammatory disorder of the blood vessels, can cause phenoconversion.
      • (lines 427-337 and added in table 2) In a prospective study in 31 patients with hepatitis C virus (HCV)–positive chronic hepatitis or cirrhosis and 30 healthy volunteers the aim was to assess the interaction between chronic liver disease and CYP2C19 genotype [40]. Metabolic ratios of omeprazole/5-hydroxy omeprazole were used as phenotypic test of CYP2C19 activity. In healthy volunteers metabolic ratios of 0.81, 1.55 and 15.5 were observed in CYP2C19 NM, IM and PM, respectively. In contrast, patients with chronic liver disease genotypically classified as CYP2C19 NM, EM and PM displayed metabolic ratios of 17.15 (21.1 fold change), 20.02 (12.4 fold change) and 26.04 (1.9 fold change), which would phenotypically classify them all as poor metabolizers of CYP2C19. These results demonstrate that chronic liver disease resulting from HCV infection can reduce CYP2C19 enzymatic activity and trigger phenoconversion cause phenoconversion in an infection- and genotype-dependent manner.

Reviewer 2 Report

It is a very interesting systematic review on the effects of phenoconversion of CYPs due to non-genetic factors.

  1. The authors correctly describe the search terms, but with them, other works that also describe the effect of phenoconversion of CYP2D6 have been left out, as is the case of two works that are discussed below.

In a study, the inhibition of debrisoquine metabolism by thioridazine was genotype dependent  (Llerena et al. 2001). Patients with CYP2D6 IM genotype were phenoconverted into PM phenotype at a lower dose (50 mg day−1 or greater) of thioridazine than those of CYP2D6 EM genotype (150 mg day−1). Previously, in another study, the inhibition of debrisoquine hydroxylation with quinidine was observed in subjects with three or more functional CYP2D6 genes (Dalén et al., 2000).

They are a couple of examples, but surely there are some more. It is suggested to add to the search other terms or to add other studies that appear in reviews such as those of references #9 or #13

References:

LLerena A, Berecz R, de la Rubia A, Fernández-Salguero P, Dorado P. Effect of thioridazine dosage on the debrisoquine hydroxylation phenotype in psychiatric patients with different CYP2D6 genotypes. Ther Drug Monit. 2001;23(6):616-620.

Dalén P, Dahl M, Andersson K, Bertilsson L. Inhibition of debrisoquine hydroxylation with quinidine in subjects with three or more functional CYP2D6 genes. Br J Clin Pharmacol. 2000;49(2):180-184.

  1. Tables 1 and 2 are mixed and the columns are not well understood. It should be stated that it means the column "type of study".

Author Response

It is a very interesting systematic review on the effects of phenoconversion of CYPs due to non-genetic factors.

  1. The authors correctly describe the search terms, but with them, other works that also describe the effect of phenoconversion of CYP2D6 have been left out, as is the case of two works that are discussed below.

In a study, the inhibition of debrisoquine metabolism by thioridazine was genotype dependent  (Llerena et al. 2001). Patients with CYP2D6 IM genotype were phenoconverted into PM phenotype at a lower dose (50 mg day−1 or greater) of thioridazine than those of CYP2D6 EM genotype (150 mg day−1). Previously, in another study, the inhibition of debrisoquine hydroxylation with quinidine was observed in subjects with three or more functional CYP2D6 genes (Dalén et al., 2000).

They are a couple of examples, but surely there are some more. It is suggested to add to the search other terms or to add other studies that appear in reviews such as those of references #9 or #13

References:

LLerena A, Berecz R, de la Rubia A, Fernández-Salguero P, Dorado P. Effect of thioridazine dosage on the debrisoquine hydroxylation phenotype in psychiatric patients with different CYP2D6 genotypes. Ther Drug Monit. 2001;23(6):616-620.

Dalén P, Dahl M, Andersson K, Bertilsson L. Inhibition of debrisoquine hydroxylation with quinidine in subjects with three or more functional CYP2D6 genes. Br J Clin Pharmacol. 2000;49(2):180-184.

    • Response: We thank the reviewer for this valuable comment regarding our search strategy. To address this comment, we have revisited our search strategy and carefully looked at the terms and papers that appear in reviews as suggested by the reviewer. As a result, we have expanded the search string with the terms intermediate, rapid and ultra-rapid metabolizer. This has resulted in the inclusion of 2 new studies. Furthermore, reexamination of the studies in the search string by comparing to the references #9 and #13 added 2 more studies. Lastly, the studies citing Llerena et al. 2001 and Dalén et al. 2000 resulted in one additional study.

We thank the reviewer for referring us to two potential phenoconversion studies. For phenoconversion resulting from concomitant medication we aimed to make a clear distinction between traditional drug-drug interaction studies, in which the genotype is not taken in consideration, and drug-drug-gene interaction studies. The exact criteria are formulated in the methods section (page 3 line 104-112). One important criteria for studies investigating phenoconversion resulting from concomitant medication was that the effect of phenoconversion should be studied for different genotype-based predicted phenotypes, as this would allow conclusions whether and how the outcome of drug-drug interactions would be modulated by the different genotypes.

The study of Llerena et al. 2001 describes phenoconversion according to our description and is therefore added to the review. The study of Dalén et al., indeed describes phenoconversion for UM subjects. However, this study does not examine other predicted phenotypes than UM, this study therefore did not match the inclusion criteria and was not included in table 1.

The following sections were added to the manuscript:

    • Revised manuscript:
      • (lines 222-237 and added in table 1) For anticoagulant drugs one paper evaluating phenoconversion was identified related to co-medication-induced phenoconversion. In a prospective study of 82 heart surgery patients (CYP2C9 *1/*1 and VKORC1 T/T) the influence of the CYP2C19 genotype on the occurrence of bleeding events during treatment with warfarin was investigated [24]. Half of the patients (n=41) were concomitantly treated with lansoprazole (a CYP2C19 inhibitor) and the other half (n=41) with rabeprazole (control group). Warfarin dose prior to concomitant treatment with a proton-pump inhibitor was based on their INR (international normalized ratio). The outcome of the study was the number of bleeding events, such as tarry stool, bleeding from colon diverticulum, conjunctival bleeding, or bleeding into the shoulder joint or surgical sites. This study showed during the 2-month follow-up that patients treated with the CYP2C19 inhibitor lansoprazole (24%) had an increased risk of developing bleeding events post-surgery compared to users of the rabeprazole (0%) that did not affect CYP2C19. Further examining this outcome for the different predicted phenotypes revealed that 40% of the CYP2C19 IM patients who concomitantly received lansoprazole developed a bleeding event, which was greater than the incidences of bleedings seen in the CYP2C19 NM (12%) and PM (22%) patients. Indicating, the distinct effect of the CYP2C19 inhibitor lansoprazole on the different CYP2C19 metabolizer phenotype groups treated with warfarin.
      • (lines 239-252 and added in table 1) For antihypertensive drugs one paper evaluating co-medication-related phenoconversion was included. In a prospective study of 16 male psychiatric patients treated with thioridazine, the influence of thioridazine withdrawal on the debrisoquine (a CYP2D6 probe drug) MR was evaluated together with the effect of CYP2D6 genotype [25]. When treated with thioridazine (150 mg/d or higher) fourteen patients (87.5%) were phenotypical CYP2D6 PM subjects (MR > 12.6). After complete withdrawal of thioridazine in ten patients, only two genotypical CYP2D6 PM patients remained phenotypical CYP2D6 PM. All of the CYP2D6 NM genotype patients returned CYP2D6 phenotypical NMs after complete withdrawal. At 50 and 100 mg of thioridazine the proportion of phenotypical CYP2D6 NM subjects decreased to 33% and 29%, respectively. Genotypic CYP2D6 IM patients receiving 50 mg of thioridazine per day, were all converted to the CYP2D6 PM metabolizer phenotype, compared to 67% of the genotypic CYP2D6 NM patients. These results show that genotypic CYP2D6 IMs are more susceptible to phenoconversion by thioridazine compared to CYP2D6 NM. Moreover, these results indicate that concomitant use of thioridazine results in phenoconversion with a dose dependent effect.
      • (lines 328-337 and added in table 1) One study examined phenoconversion by concomitant medication in patients using NSAIDs. This prospective cross-over study examined in 22 healthy volunteers the consequences of 7-days concomitant treatment with the CYP2C9 inhibitor fluconazole for the metabolism of the CYP2C9 substrate flurbiprofen [31]. Co-administration of 200 mg fluconazole increased the AUC of flurbiprofen in the 11 CYP2C9 predicted NM to an extent that was typical for CYP2C9 IM. Higher dosages of fluconazole (400 mg) changed the AUC even further to an extent that was indicative for the conversion into CYP2C9 PM. On the other hand, both dosages of fluconazole reduced the AUC of flurbiprofen in the 8 predicted IM subjects to levels that were representative for CYP2C9 PM subjects. These results indicate that concomitant use of fluconazole can cause phenoconversion in a dose- and genotype-dependent manner.
      • (lines 351-366 and added in table 1) Two papers investigated phenoconversion in patients using anti-estrogenic drugs, one paper due to concomitant medication and one paper due to vitamin D exposure. In a prospective cohort study 158 breast cancer patients were treated with tamoxifen and received concomitant treatment with CYP2D6 inhibitors [33]. The study aim was to show that concomitant treatment with CYP2D6 inhibitors reduces the endoxifen plasma concentration. Out of the 158 patients, 17 patients used potent CYP2D6 inhibitors (paroxetine and fluoxetine) and 25 patients used weak CYP2D6 inhibitors (sertraline, citalopram, celecoxib, diphenydramine, and chlorpheniramine). Eighty-five patients not taking any concomitant CYP2D6 medication were used as control group. With the exception of CYP2D6 UM, all patients receiving concomitant treatment with a strong CYP2D6 inhibitor were phenoconverted to the CYP2D6 PM metabolizer phenotype. The CYP2D6 UM patients treated with weak or potent CYP2D6 inhibitors did show lower endoxifen plasma concentrations, comparable to the CYP2D6 NM or IM metabolizer phenotype. Concomitant treatment with a weak inhibitor CYP2D6 inhibitor resulted for patients with the CYP2D6 NM genotype in conversion to the CYP2D6 IM metabolizer phenotype and CYP2D6 IM patients were converted to the CYP2D6 PM metabolizer phenotype. This, again, shows that in addition to the genotype also the potency of the inhibitor plays an important role in the occurrence of phenoconversion.
      • (lines 421-427 and added in table 2) In a prospective study in 52 patients with Behçet disease and 96 healthy volunteers the aim was to investigate the influence of Behçet disease and CYP2C9 genotype on the activity of CYP2C9, which was phenotypically determined by measuring the metabolic ratio of losartan [39]. The 31 patients that were genotypically classified as CYP2C9 NM had a mean metabolic ratio that was comparable to the observed metabolic ratio of losartan in CYP2C9 IM healthy volunteers. These results indicate that Behçet disease, an systemic inflammatory disorder of the blood vessels, can cause phenoconversion.
      • (lines 427-337 and added in table 2) In a prospective study in 31 patients with hepatitis C virus (HCV)–positive chronic hepatitis or cirrhosis and 30 healthy volunteers the aim was to assess the interaction between chronic liver disease and CYP2C19 genotype [40]. Metabolic ratios of omeprazole/5-hydroxy omeprazole were used as phenotypic test of CYP2C19 activity. In healthy volunteers metabolic ratios of 0.81, 1.55 and 15.5 were observed in CYP2C19 NM, IM and PM, respectively. In contrast, patients with chronic liver disease genotypically classified as CYP2C19 NM, EM and PM displayed metabolic ratios of 17.15 (21.1 fold change), 20.02 (12.4 fold change) and 26.04 (1.9 fold change), which would phenotypically classify them all as poor metabolizers of CYP2C19. These results demonstrate that chronic liver disease resulting from HCV infection can reduce CYP2C19 enzymatic activity and trigger phenoconversion cause phenoconversion in an infection- and genotype-dependent manner.
  1. Tables 1 and 2 are mixed and the columns are not well understood. It should be stated that it means the column "type of study".
    • Response: We thank the reviewer for this comment, the tables are used to provide an overview and need to be clear, therefore the order of columns in table 1 is adjusted to read more similar to table 2. And the description of the “type of study” is added in the table title.
    • Revised manuscript:
      • (lines 166-168 and lines 177-179 and table 1 column order) Type of study refers to “designed with the objective to study phenoconversion” (type 1) or “not specifically designed to study phenoconversion , did not categorize patients into PM, IM, NM, RM or UM phenotype” (type 2).

Reviewer 3 Report

The authors/investigators present a review of the literature relative to phenoconversion. This is an important contribution to the literature to put drug-drug-gene interactions and pathophysiology-drug-gene interactions into perspective. I have only minor comments:

  1. Line 43 - Should rapid metabolizers (RM) be included as a fifth metabolizer phenotype? Certainly for CYP2C19, this phenotype is addressed in a number of guidelines and the RM phenotype is being identified and included for more distinct categorization.
  2. Lines 77-80 - It may be valuable here to mention weak, moderate, and strong inhibitors to put those in context, i.e., If John was a NM receiving a strong inhibitor, what would the potential consequences be? https://drug-interactions.medicine.iu.edu/MainTable.aspx
  3. Table 1 - "36 PM phenytoin/carbamazeine users" should read "36 PM phenytoin/carbamazepine users" i.e., carbamazepine typographical error.
  4. Line 162 - "In a retrospective study of 28 patients with Dravet syndrome receiving valproate and concomitant stiripentol it was aimed to elucidate the mechanism underlying the increase in serum valproate concentration produced by concomitant stiripentol therapy"

    Consider rewording. "it was" does not read well.

    Maybe "A retrospective study of 28 patients with Dravet syndrome receiving valproate and concomitant stiripentol was undertaken to elucidate the mechanism underlying the increase in serum valproate concentration produced by concomitant stiripentol therapy"?

  5. Line 165 - topiramaat, not toiromat?
  6. Lines 190-191 - Reword to "decreased by 80% in CYP2D6 NM subjects 190 and by 93% in CYP2D6 IM subjects."?

    Decreased with does not convey the meaning of decreased by.

  7. Line 194 - For "Two attributed to co-medication-induced phenoconversion and one investigating smoking-induced phenoconversion", consider "Two attributed to co-medication-related phenoconversion and one investigating smoking-related phenoconversion"

    The reasoning here is that using the relationship in terms of "co-medication-induced phenoconversion" may confuse the reader as the word "induced" may be thought of relative to metabolism, i.e., induction, when in fact you are really discussing CYP2D6 inhibitors, in part.

  8. Lines 277-278 - Really do not need the last sentence here as the section header describes this.
  9. Line 352 - Add space between "CYP2D6" and "enzyme"
  10. Line 372 - Can the authors comment on liver disease? A number of papers have described altered metabolic ratios due to different liver diseases. Maybe these did not meet the criteria set, however, I thought at least one may. CPT 2006;80(3):235-45. doi: 10.1016/j.clpt.2006.05.006. Just checking.

Author Response

The authors/investigators present a review of the literature relative to phenoconversion. This is an important contribution to the literature to put drug-drug-gene interactions and pathophysiology-drug-gene interactions into perspective. I have only minor comments:

  • Line 43 - Should rapid metabolizers (RM) be included as a fifth metabolizer phenotype? Certainly for CYP2C19, this phenotype is addressed in a number of guidelines and the RM phenotype is being identified and included for more distinct categorization.
    • Response: We thank the reviewer for this comment, we focused on the DPWG guidelines which do not include RMs. However, we agree with the reviewer that in the CPIC guidelines there are different recommendations for CYP2C19 RMs. Therefore we added CYP2C19 RMs as a category of metabolizer phenotypes and included this in the adapted search string as well.
  • Original manuscript:
      • (lines 42-45) Historically, based on observed variability in pharmacokinetics, genetic variants are categorized into 4 different predicted metabolizer phenotypes; normal metabolizers (NM), ultra-rapid metabolizers (UM), intermediate metabolizers (IM) and poor metabolizers (PM).
    • Revised manuscript:
      • (lines 46-49) Historically, based on observed variability in pharmacokinetics, genetic variants are categorized into 5 different predicted metabolizer phenotypes; normal metabolizers (NM), ultra-rapid metabolizers (UM), rapid metabolizers (RM), intermediate metabolizers (IM) and poor metabolizers (PM).
  • Lines 77-80 - It may be valuable here to mention weak, moderate, and strong inhibitors to put those in context, i.e., If John was a NM receiving a strong inhibitor, what would the potential consequences be? https://drug-interactions.medicine.iu.edu/MainTable.aspx
    • Response: We thank the reviewer for this comment, we added the case example as an explanation of phenoconversion. Nevertheless, we do agree that there are potential different consequences for weak and moderate inhibitors. To address the reviewers comment, we emphasized the effect of weak/moderate/strong inhibitors more throughout the manuscript.
  • Original manuscript:
      • (lines 408-409) This finding may be particularly important for the concomitant use of weak or moderate CYP450 inhibitors and warrant further investigations.
    • Revised manuscript:
      • (lines 239-252 and added in table 1) For antihypertensive drugs one paper evaluating co-medication-related phenoconversion was included. In a prospective study of 16 male psychiatric patients treated with thioridazine, the influence of thioridazine withdrawal on the debrisoquine (a CYP2D6 probe drug) MR was evaluated together with the effect of CYP2D6 genotype [25]. When treated with thioridazine (150 mg/d or higher) fourteen patients (87.5%) were phenotypical CYP2D6 PM subjects (MR > 12.6). After complete withdrawal of thioridazine in ten patients, only two genotypical CYP2D6 PM patients remained phenotypical CYP2D6 PM. All of the CYP2D6 NM genotype patients returned CYP2D6 phenotypical NMs after complete withdrawal. At 50 and 100 mg of thioridazine the proportion of phenotypical CYP2D6 NM subjects decreased to 33% and 29%, respectively. Genotypic CYP2D6 IM patients receiving 50 mg of thioridazine per day, were all converted to the CYP2D6 PM metabolizer phenotype, compared to 67% of the genotypic CYP2D6 NM patients. These results show that genotypic CYP2D6 IMs are more susceptible to phenoconversion by thioridazine compared to CYP2D6 NM. Moreover, these results indicate that concomitant use of thioridazine results in phenoconversion with a dose dependent effect.
      • (lines 306-310) Almost all of the heterozygous CYP2D6 NM subjects were phenoconverted due to the strong inhibitor paroxetine. For the moderate inhibitor duloxetine this was only 71%. This difference was even bigger in homozygous CYP2D6 IM subjects, 56% vs 25% for paroxetine and duloxetine, respectively. These results show that the strength of the inhibitor (weak/moderate/strong) important also influences the occurrence of phenoconversion.
      • (lines 328-337 and added in table 1) One study examined phenoconversion by concomitant medication in patients using NSAIDs. This prospective cross-over study examined in 22 healthy volunteers the consequences of 7-days concomitant treatment with the CYP2C9 inhibitor fluconazole for the metabolism of the CYP2C9 substrate flurbiprofen [31]. Co-administration of 200 mg fluconazole increased the AUC of flurbiprofen in the 11 CYP2C9 predicted NM to an extent that was typical for CYP2C9 IM. Higher dosages of fluconazole (400 mg) changed the AUC even further to an extent that was indicative for the conversion into CYP2C9 PM. On the other hand, both dosages of fluconazole reduced the AUC of flurbiprofen in the 8 predicted IM subjects to levels that were representative for CYP2C9 PM subjects. These results indicate that concomitant use of fluconazole can cause phenoconversion in a dose- and genotype-dependent manner.
      • (lines 534-537) This finding may be particularly important for the concomitant use of weak or moderate CYP450 inhibitors, which hypothetically will have the strongest effect on phenoconversion in IM subjects and warrant further investigations.
  • Table 1 - "36 PM phenytoin/carbamazeine users" should read "36 PM phenytoin/carbamazepine users" i.e., carbamazepine typographical error.
    • Response: We thank the reviewer for spotting this typo. We have corrected it.
  • Original manuscript:
      • (table 1) 36 PM phenytoin/carbamazeine users
    • Revised manuscript:
      • (table 1) 36 PM phenytoin/carbamazepine users
  • Line 162 - "In a retrospective study of 28 patients with Dravet syndrome receiving valproate and concomitant stiripentol it was aimed to elucidate the mechanism underlying the increase in serum valproate concentration produced by concomitant stiripentol therapy"

Consider rewording. "it was" does not read well.

Maybe "A retrospective study of 28 patients with Dravet syndrome receiving valproate and concomitant stiripentol was undertaken to elucidate the mechanism underlying the increase in serum valproate concentration produced by concomitant stiripentol therapy"?

    • Response: We thank the reviewer for this correction. We do agree that this reads better, so we replaced the sentence by the advised phrase.
  • Original manuscript:
      • (lines 162-164) In a retrospective study of 28 patients with Dravet syndrome receiving valproate and concomitant stiripentol it was aimed to elucidate the mechanism underlying the increase in serum valproate concentration produced by concomitant stiripentol therapy
    • Revised manuscript:
      • (lines 193-195) A retrospective study of 28 patients with Dravet syndrome receiving valproate and concomitant stiripentol was undertaken to elucidate the mechanism underlying the increase in serum valproate concentration produced by concomitant stiripentol therapy
  • Line 165 - topiramaat, not toiromat?
    • Response: We thank the reviewer for spotting this typo. We have corrected it.
  • Original manuscript:
      • (lines 164-167) In patients not concomitantly treated with topiromaat, it was found that the increase in valproate serum concentration was larger for CYP2C19 NM subjects compared to CYP2C19 PMs, suggesting that CYP2C19 NM subjects are more susceptible to the effects of CYP2C19 inhibition.
    • Revised manuscript:
      • (lines 199-201) It was found that the increase in valproate serum concentration was larger for CYP2C19 NM subjects compared to CYP2C19 PMs, suggesting that CYP2C19 NM subjects are more susceptible to the effects of CYP2C19 inhibition.
  • Lines 190-191 - Reword to "decreased by 80% in CYP2D6 NM subjects 190 and by 93% in CYP2D6 IM subjects."?

Decreased with does not convey the meaning of decreased by.

    • Response: We thank the reviewer for this correction. We do agree that this is a better phrase, so we replaced “with” by “by”.
  • Original manuscript:
      • (lines 189-191) The authors reported that the oral clearance of tolterodine decreased with 80% in CYP2D6 NM subjects and with 93% in CYP2D6 IM subjects.
    • Revised manuscript:
      • (lines 256-258) The authors reported that the oral clearance of tolterodine decreased by 80% in CYP2D6 NM subjects and with 93% in CYP2D6 IM subjects.
  • Line 194 - For "Two attributed to co-medication-induced phenoconversion and one investigating smoking-induced phenoconversion", consider "Two attributed to co-medication-related phenoconversion and one investigating smoking-related phenoconversion"

The reasoning here is that using the relationship in terms of "co-medication-induced phenoconversion" may confuse the reader as the word "induced" may be thought of relative to metabolism, i.e., induction, when in fact you are really discussing CYP2D6 inhibitors, in part.

    • Response: We thank the reviewer for this correction. We do agree that this is more precise, so we replaced the sentence by the advised phrase.
  • Original manuscript:
      • (lines 194-195) Two attributed to co-medication-induced phenoconversion and one investigating smoking-induced phenoconversion.
    • Revised manuscript:
      • (lines 262-264) Three attributed to co-medication-related phenoconversion and one investigating smoking-related phenoconversion
  • Lines 277-278 - Really do not need the last sentence here as the section header describes this.
    • Response: We thank the reviewer for this comment. We do agree, so corrected it by deleting the sentence.
  • Original manuscript:
      • (lines 277-278) In the next section studies assessing patient- and disease-related factors in relation to phenoconversion will be described.
  • Line 352 - Add space between "CYP2D6" and "enzyme"
    • Response: We thank the reviewer for spotting this typo. We have corrected it.
  • Original manuscript:
      • (lines 352-353) CYP2D6enzymatic activity was increased in genotypic subjects with a CYP2D6 NM and UM genotype and reduced in subjects with a CYP2D6 PM genotype.
    • Revised manuscript:
      • (lines 477-478) CYP2D6 enzymatic activity was increased in genotypic subjects with a CYP2D6 NM and UM genotype and reduced in subjects with a CYP2D6 PM genotype.
  • Line 372 - Can the authors comment on liver disease? A number of papers have described altered metabolic ratios due to different liver diseases. Maybe these did not meet the criteria set, however, I thought at least one may. CPT 2006;80(3):235-45. doi: 10.1016/j.clpt.2006.05.006. Just checking.
    • Response: We thank the reviewer for this question. We expected to find an effect of liver diseases on phenoconversion, therefore we included it as a search term in our search string. One study in patients with chronic liver disease was identified (page 5 line 396-406). Nevertheless, in our opinion the phenoconversion that occurred can be attributed to the presence of inflammation which caused the liver disease.

We thank the reviewer for bringing the study by Frye et al. under our attention. We used specific criteria for inclusion of studies examining phenoconversion, these criteria are formulated in the methodology (page 3 line 104-112). In our opinion it was necessary to assess the genotype, because genotype-based predicted phenotypes, as this would allow conclusions whether and how the outcome of the interaction with the gene would be modulated and result in a different phenotype. Although the study by Frye et al. demonstrated phenoconversion, this study did not assess the genotype. Consequently, the study was not included in our systematic review, as this study did not meet our inclusion criteria.

Nevertheless, we do agree that liver diseases is an important point to consider for phenoconversion. Therefore, we emphasized the effect of liver diseases more in the discussion.

    • Revised manuscript:
      • (lines 544-546) . Liver diseases has also been suggested as source of phenoconversion. However, while a number of studies did show changes in plasma levels due to liver diseases, no adequate genotype assessments were available in those studies [59-61].

[59] Frye, R.F., et al., Liver disease selectively modulates cytochrome P450–mediated metabolism. Clinical Pharmacology & Therapeutics, 2006. 80(3): p. 235-245.

[60] Pageaux, G.P., et al., Pharmacokinetics of sabeluzole and dextromethorphan oxidation capacity in patients with severe hepatic dysfunction and healthy volunteers. Br J Clin Pharmacol, 2001. 51(2): p. 164-8.

[61] Shao, J.G., et al., Blood concentration of pantoprazole sodium is significantly high in hepatogenic peptic ulcer patients, especially those with a poor CYP2C19 metabolism. J Dig Dis, 2009. 10(1): p. 55-60.
